# Gradient-Weight Alignment as a Train-Time Proxy for Generalization in Classification Tasks

**Florian A. Hölzl, Daniel Rueckert, Georgios Kaissis**
Institute for Artifical Intelligence in Medicine
Technical University of Munich
{florian.hoelzl, daniel.rueckert, g.kaissis}@tum.de

## Abstract

Robust validation metrics remain essential in contemporary deep learning, not only to detect overfitting and poor generalization, but also to monitor training dynamics. In the supervised classification setting, we investigate whether interactions between training data and model weights can yield such a metric that both tracks generalization during training and attributes performance to individual training samples. We introduce *Gradient-Weight Alignment* (GWA), quantifying the coherence between per-sample gradients and model weights. We show that effective learning corresponds to coherent alignment, while misalignment indicates deteriorating generalization. GWA is efficiently computable during training and reflects both sample-specific contributions and dataset-wide learning dynamics. Extensive experiments show that GWA accurately predicts optimal early stopping, enables principled model comparisons, and identifies influential training samples, providing a validation-set-free approach for model analysis directly from the training data.

## 1 Introduction

Despite the recent surge in self-supervised learning, optimization with cross-entropy loss remains dominant in modern deep learning for both supervised training and fine-tuning. Its enduring popularity stems from its strong label-based learning signal and implicit modeling of predictions as probabilities. However, this very strength – relying on maximum likelihood estimation – introduces vulnerabilities to overconfidence and sensitivity to noise in both input data and labels. Diagnosing these issues typically relies on hold-out validation sets, which operate under the assumption of independent and identically distributed (*i.i.d.*) data. While effective, this standard approach necessitates labeled data that is rendered unavailable for training and offers limited insight into how these issues can be attributed to training set samples. This motivates a critical question: *can we effectively assess model generalization and diagnose potential problems solely using information available during training?*

Prior work indicates that robust generalization emerges when all training samples contribute coherently towards a shared learning goal, that is, their gradients are directionally well aligned [1, 2]. Conversely, conflicting or misaligned gradient directions indicate a potential failure to generalize. However, computing pairwise gradient alignment is highly memory-intensive and is not an inherently distributional quantity; in other words, the average gradient alignment over the dataset provides minimal insight into individual samples' contribution to training. In this work, we thus turn to the alignment between per-sample gradients and the model weights as a measure for estimating key training dynamics and predicting generalization in training with cross-entropy. This quantity, which we refer to as Gradient-Weight Alignment (GWA), as well as its distribution, capture the degree of gradient coherence across the dataset and allow linking model performance directly to the individual underlying input-label pairs (Fig. 1). Moreover, we show that GWA can be efficiently estimated during training (online), providing insights into training dynamics even during large-scale optimization with negligible overhead. We argue through extensive empirical evaluation that GWA accurately

39th Conference on Neural Information Processing Systems (NeurIPS 2025).

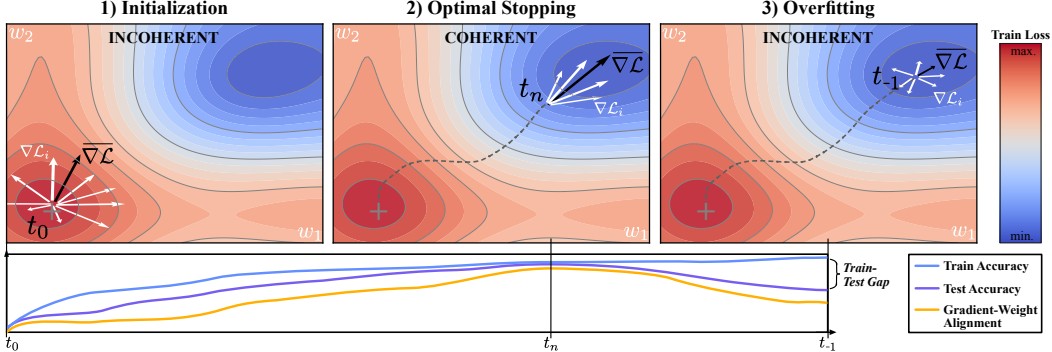

Figure 1: Gradient alignment among individual samples $\nabla\mathcal{L}_i$ as well as the model weights varies during training, with coherent per-sample gradient direction reflecting generalization. Line plots illustrate how GWA captures gradient coherence and model performance at different time points $t$.

identifies the point in training beyond which further training steps yield diminishing returns in terms of generalization performance, even rendering held-out validation sets redundant. Our contributions can be summarized as follows:

- We introduce GWA as a novel proxy for generalization performance during training - effectively replacing the need for withholding a separate validation set.
- GWA reveals the influence of individual training samples on optimization, providing a powerful diagnostic tool for understanding data quality issues like outliers and label errors.
- We show GWA scales to large-scale training and finetuning, and remains robust under input/label noise, offering a superior criterion for choosing models deployed in practice.

## 2   Related Work

Quantifying the generalization gap - understanding *how well* a model performs on unseen data - is a central challenge in deep learning [3]. Early approaches without hold-out data focused on estimating this gap using the curvature of the loss function, offering insights into sample influence. However, computing second-order derivatives is computationally expensive [4, 5, 6, 7], and curvature can be unstable during training and sensitive to hyperparameters [8, 9, 10]. More recently, influence functions[11] and sub-sampling estimators [12, 13] have emerged to compute per-sample influence at the end of training without second-order derivatives. Our work differs by focusing on an efficient train-time estimator of generalization for online assessment rather than post-training influence analysis.

We contend that comprehensively evaluating and characterizing model dynamics is crucial for understanding optimization and its challenges - such as potential overfitting. Prior work has observed that Stochastic Gradient Descent (SGD) exhibits a "simplicity bias", initially learning simple patterns before fitting increasingly complex functions [14, 15, 16, 17, 18, 19]. While we build on these insights, we move beyond characterizing what is learned to quantifying when further learning yields diminishing returns for generalization – a relatively underexplored area outside the noisy label regime [20, 21]. Most notably from this area, we compare against the recent *LabelWave* [22, 23] which quantifies model prediction changes to determine a suitable early stopping point. Prediction-change-focused methods like *LabelWave* are designed primarily for early stopping in noisy optimization, but offer limited insight into underlying training dynamics. In contrast, our GWA directly reflects these dynamics via a sample-level measure.

Quantifying the coherence of per-sample gradients captures training dynamics such as faster convergence and improved generalization [24, 1, 25, 26, 27]. As illustrated in Fig. 1, this coherence is captured by the *direction* of the gradients, a signal distinct from their magnitude. However, computing coherence in the aforementioned approaches requires the gradients of all training samples to be stored in memory – an impractical limitation for large datasets – and often provides only aggregate statistics, obscuring valuable per-sample insights. Gradient Disparity (GD) [28] provides a more efficient approach for $k$-fold cross validation but has not been shown to work in large scale settings. Our

approach is scalable and addresses these limitations by leveraging the model weights as a reference vector. Theoretical work demonstrates that under ideal conditions -perfectly classifiable data- weights converge in direction, *i.e.*, maintain a specific direction while the gradient aligns with the weights' direction [29, 2]. While the impact of noise in this case is still unexplored, some levels of noise can be beneficial [30, 31], and recent empirical research affirms directional stability even with stochasticity by analyzing batch gradient direction relative to the optimal weights [32]. Our proposed GWA specifically focuses on quantifying alignment in the presence of realistic data variance. Moving from pairwise and aggregate gradient statistics to GWA is not only more efficient but allows us to link individual samples to generalization performance in stochastic optimization.

In the following, we propose an efficient computation strategy of per-sample *gradient-weight alignment* and leverage the properties of the resulting alignment distribution as a novel measure of generalization and sample-level influence.

## 3  Gradient-Weight Alignment

We begin by introducing GWA, the key quantity studied in our work. GWA is inspired by theoretical work on the directional convergence of model weights learned by gradient flow when minimizing the cross-entropy loss [2]. Intuitively, the fundamental idea of this work is that, for perfectly classifiable data, the weights not only converge *in direction*, but moreover the corresponding gradients converge *in direction to the weights*, *i.e.*, the gradient and weights align.

**Motivation**    The central hypothesis of our work is therefore that *in practice* the alignment between *per-sample* gradients and model weights can be leveraged to capture the convergence and consequent *generalization* of a model. Differences in per-sample alignment link the model's performance to the individual data samples. This inherently requires studying GWA through two complementary lenses: (1) the per-sample alignment scores, which quantify how well individual samples of a real-life, noisy dataset are represented in the general optimization trajectory and (2) a distributional measure of the alignment scores across the dataset, which reflects the degree to which dataset properties (*e.g.*, data diversity, variance, etc.) impact the resulting generalization across training iterations. We first formally define GWA and its related quantities.

**Definition 1 (Per-Sample Alignment)** *Let $\mathbf{g}_t(x_i) = -\nabla_{\mathbf{w}}\mathcal{L}(\mathbf{w}_t, x_i)$ denote the negative gradient of the loss function with respect to the model weights $\mathbf{w}_T$ at a single input-label pair $(x_i, y_i)$, where we drop $y_i$ for brevity, and at epoch $T$. Then, the per-sample alignment score is defined as:*

$$\gamma(x_i, \mathbf{w}_T) = \cos\sim\left(\mathbf{g}_T(x_i), \mathbf{w}_T\right) = \frac{\mathbf{g}_T(x_i) \cdot \mathbf{w}_T}{\|\mathbf{g}_T(x_i)\|\|\mathbf{w}_T\|}. \tag{1}$$

The per-sample alignment score $\gamma(x_i, \mathbf{w}_T)$ effectively measures how well the model weights align with the "learning direction" for that specific instance. Intuitively, a higher score suggests that the model is more efficiently incorporating information from that sample into its weight updates. The theoretical justification for the per-sample alignment score is derived from the fact that, under ideal conditions of perfectly classifiable data, the predicted probabilities asymptotically approach the target and alignment should satisfy $\mathbb{E}_i[\gamma(x_i, \mathbf{w}_T)] \to 1$ for large enough $T$ [2]; in other words, the gradient would consistently point in the same direction as the model weights. While perfect convergence is rarely observed with real-world datasets due to inherent noise and complexity, this theoretical limit provides a valuable intuition: *a model with better generalization performance should exhibit higher average $\gamma(x_i, \mathbf{w}_T)$*. Conversely, consistently low alignment scores, *i.e.*, updates orthogonal to the direction of the model weights, can signal issues like noisy labels or learning non-general, sample specific information and potential overfitting.

**Definition 2 (Gradient-Weight Alignment)** *Let $\mathcal{G}_T \coloneqq \{\gamma(x_i, \mathbf{w}_T)\}_{i=0}^N, \gamma(x_i, \mathbf{w}_T) \in [-1, 1]$ be the set of per-sample alignment scores at epoch $T$. Moreover, let $\mathcal{A}_T$ denote the empirical distribution of the values of $\mathcal{G}_T$ and $M_T^{(k)}$ its $k^{th}$ moment. Then, the gradient-weight alignment (GWA) at epoch $T$ is defined as the excess-kurtosis-corrected expectation of $\mathcal{A}_T$:*

$$\mathrm{GWA}_T = \frac{\mathbb{E}_i[\mathcal{A}_T]}{\mathrm{Kurt}_i[\mathcal{A}_T] + \beta} = \frac{M_T^{(1)}}{M_T^{(4)}/\left(M_T^{(2)}\right)^2 - 3 + \beta}. \tag{2}$$

Above, $\beta$ is added to ensure non-negativity of the denominator. We choose $\beta = 1.2$ to offset the excess kurtosis $\mathrm{Kurt}$ of the uniform distribution over $[-1, 1]$, which is a limiting case never to be expected in practice.

Unlike theoretical work that assumes perfectly classifiable data, real-world datasets can exhibit high degrees of variance and noise. The distribution of per-sample alignment scores $\gamma(x_i, \mathbf{w}_T)$ provides valuable insights into the quality of learning – a coherent alignment distribution (*i.e.*, high GWA) indicates consistent gradient directions across samples and thus effective generalization. At this point, one may wonder why GWA considers not only the mean but also the kurtosis of the alignment distribution. The theoretical motivation for incorporating the kurtosis, a measure of tailedness of the distribution, in other words, of the impact of rare samples, is derived from the long-tail theory of deep learning [12, 13]. This line of work demonstrates that, in natural image tasks, rare/atypical samples have an outsized influence on the model. Distributions in which rare samples exert high influence are commonly referred to as "heavy-tailed" distributions and have high kurtosis. [1] In other words, a high kurtosis value in the GWA denominator intuitively indicates a large proportion of samples with disproportionate influence on the overall alignment, thus signaling potentially problematic learning patterns by diminishing the GWA value.

Note that we choose the value of the $\beta$ factor such that the kurtosis has only minimal impact on GWA when the distribution is a (truncated) Gaussian ($\mathrm{Kurt}_i[\mathcal{A}_T] \approx 0$). For more concentrated (platykurtic) distributions with lower kurtosis (*e.g.*, approaching a uniform), the GWA value increases, and for distributions with high kurtosis (leptokurtic, *e.g.*, Laplace) the GWA value decreases. We will use the term *(highly) coherent* synonymously with high GWA. As we show experimentally below, tracking GWA over time reveals critical training dynamics related to overfitting vs. generalization.

**Scalable Estimator**    Capturing the per-sample alignment properties during training on large datasets necessitates a scalable GWA estimation approach. However, directly computing $\gamma(x_i, \mathbf{w}_T)$ for all samples at each timestep is computationally prohibitive. We address this by leveraging inherent properties of supervised classification models and GWA to design a lightweight estimator with minimal overhead that can be run online even for very large models and datasets.

The first primary obstacle to GWA estimation lies in the sensitivity of the cosine similarity term in Eq. (1) to high dimensionality of the arguments. Recall that the expected cosine similarity between vectors with independent noise components diminishes rapidly with increasing dimensions. Moreover, computing full network gradients at every step introduces significant implementation and computational overhead. To mitigate both aforementioned issues, we exploit the fact that classification fundamentally operates on *latent representations*. A deep classifier's primary goal is to learn a representation that is linearly separable by its final layer, with the last layer offering the most direct signal of the learned task [33]. In fact, for our case, including earlier layers degrades the estimator, with gradients in shallower layers being significantly more unstable, as shown in work such as [34]. Consequently, we propose estimating per-sample gradients using only the final layer's weights - *i.e.*, without materializing the full model's gradient. This transforms gradient computation into an efficient matrix multiplication $\mathbf{g}_T(x_i) = -z_i \cdot (\hat{y}_i - y_i)^\top$, with latent representation $z_i$, logits $\hat{y}_i$, target $y_i$, and the weights of the linear head, based on the technique proposed in [35] and shown in [36].

A second challenge arises in determining *when* to measure per-sample alignment. Computing it for every sample at each timestep incurs substantial overhead despite the efficiency improvements outlined above. We address this by proposing a computationally efficient estimator of GWA: intuitively, instead of recomputing alignment scores across all samples at a fixed gradient update step, we compute them over all gradient update steps of one epoch as follows. Let $T$ denote the current epoch, $b$ the batch size such that $K = N/b$ is the number of minibatches per epoch and $N$ is the total number of samples in the dataset. Within an epoch, let $t \in \{0, 1, \ldots, K-1\}$ index the steps per epoch and let $\mathbf{w}_{T,t}$ denote the model weights at the beginning of step $t$ of epoch $T$. Let $\mathcal{B}_{T,t} := \{(x_i, y_i) \mid i = tb+1, \ldots, (t+1)b\}$ denote the batch for step $t$. Then, we estimate the $k^{\mathrm{th}}$

---

[1] Note that the term "heavy-tailed" is formally inapplicable to distributions supported on bounded intervals, but the intuition conveyed is still valid and we thus retain this terminology.

central moment of $\mathcal{A}_T$, $\hat{M}_T^{(k)}$ as follows:

$$\hat{M}_T^{(k)} = \frac{1}{N} \sum_{t=0}^{K-1} \sum_{x_i \in \mathcal{B}_{T,t}} \left( \gamma\left(x_i, \mathbf{w}_{T,t}\right) - \hat{M}_T^{(1)} \right)^k, \tag{3}$$

where $\hat{M}_T^{(1)}$ (recursively) estimates the first central moment (mean). Note that above, the model weights are indexed twice because they change after each gradient update. The estimator of GWA is then derived by replacing the true central moments of $\mathcal{A}_T$ with the estimated central moments in the RHS of Eq. (2). Under the mild assumption of finite empirical moments and a small-enough learning rate, this estimator furnishes an extremely computationally efficient technique to monitor the alignment distribution during training even for very large models and datasets. In essence, estimating GWA in this way reduces to a single forward pass through the network's linear classifier. Algorithm 1 summarizes our approach for estimating GWA incorporating the aforementioned techniques.

---

**Algorithm 1** Estimation of $\mathrm{GWA}_T$

---

**Require:** Total per-epoch iterations $K$, batch size $b$, learning rate $\eta_t$, classifier weights $\mathbf{w}_{T,0}$
 1: **for** each iteration $t = 0, \ldots, K-1$ **do**
 2:     Sample a minibatch $\mathcal{B}_{T,t}$ of size $b$ from the dataset.
 3:     Compute standard forward pass with $\mathcal{B}_{T,t}$.
 4:     **for** each input-label pair $(x_i, y_i)$ in $\mathcal{B}_{T,t}$ **do**
 5:         Compute per-sample loss $\mathcal{L}(\mathbf{w}_t, x_i, y_i)$, softmax logits $\hat{y}_i$, and latents $z_i$.
 6:         Compute gradients of linear head in closed form: $\mathbf{g}_t(x_i) = -z_i \cdot (\hat{y}_i - y_i)^\top$
 7:         Compute and store per-sample alignment: $\gamma(x_i, \mathbf{w}_{T,t}) = \frac{\mathbf{g}_{T,t}(x_i) \cdot \mathbf{w}_{T,t}}{\|\mathbf{g}_{T,t}(x_i)\| \cdot \|\mathbf{w}_{T,t}\|}$.
 8:     **end for**
 9:     Update model with minibatch gradient based on step 3.
10: **end for**
11: **Output** $\mathrm{GWA}_T$ estimated according to Eq. (2) on per-sample alignments stored in step 7.

---

In the following sections, we demonstrate how GWA can be used to predict optimal early stopping, compare model performance across runs, and identify influential training samples – all without relying on validation sets.

## 4  Evaluation

Our primary comparison to empirically evaluate GWA is against standard validation set-based early stopping, the most common practice for evaluating generalization performance during training. To ensure reproducibility and broad applicability, we conduct experiments on ConvNeXt-Femto [37, 38] and ViT/S-16 [39, 40] architectures — both popular choices offering a balance between computational efficiency and accuracy. We leverage established public benchmarks to aid reproducibility, including ImageNet-1k (using the standard validation set for testing), ImageNet-V2 [41], and ImageNet-ReaL [42] as well as CIFAR-10 and its noisy variant CIFAR-10-N [43] to systematically assess performance under varying levels of label noise and to detect overfitting. Our fine-tuning experiments are conducted on a ViT/B-16 model pre-trained on ImageNet-21k [44].

Validation sets are created via a standard train/val split of the original validation data (*e.g.*, 90% training, 10% validation). If no test sets exist, the official validation sets are used as hold-out test sets and are referred as such in the following. All models are trained for a fixed number of optimization steps, *i.e.*, with the same compute budget regardless of training set size. Beyond label noise evaluation, we assess the robustness of models selected using different early stopping criteria on CIFAR-C and ImageNet-C [45], employing realistic input perturbations consistent with our other experiments.

When training a ViT/S-16 implemented in JAX on ImageNet-1k with a single NVIDIA RTX A6000, GWA adds $\approx 2.5$sec to the per-epoch wall-clock time (on average 1861 images/s with GWA vs. 1867 images/s without GWA for $224^2$px). This is more efficient than evaluating a 1% validation set (16sec overhead for one iteration). Computing the closed-form gradients and the cosine similarity requires $\approx 0.003$GFLOPs compared to 4.6 GFLOPs of a single forward pass with a ViT/S-16. Peak GPU memory when using active deallocation in this setting is 25.11GB with or without GWA (no

difference). Thus, GWA has minimal overhead. Detailed hyperparameters used for all approaches are provided in Appendix C. An open-source implementation of our approach in JAX and PyTorch can be found under https://github.com/hlzl/gwa.

In the following, we will demonstrate GWA's capability to predict optimal early stopping, compare models across runs and provide insights into the underlying training dynamics and the corresponding influence of individual samples – effectively replacing traditional validation strategies.

## 4.1 Early Stopping and Training Dynamics

Table 1: GWA matches or outperforms most validation metrics when used as an early-stopping criterion. Top-1 test accuracy achieved by ViT/S-16 and ConvNeXt trained from scratch on CIFAR-10(-N) (*with varying noise percentages*), and ImageNet-1k using different early stopping strategies (averaged across 3 runs, min-max range below in gray). Performances are reported as difference to baseline validation set with $90/10\%$ train/val split and compared to validation sets $99/1\%$ split, prediction changes measured by LabelWave, Gradient Disparity (GD), and our proposed GWA without validation set. Best in **bold**, second best underlined.

| | Test Accuracy [%] ($\triangle$ min–max) | | | | | | | | | | | |
| | ViT | | | | | | ConvNeXt | | | | | |
| | CIFAR-10 [label noise %] | | | ImageNet-1k | | | CIFAR-10 [label noise %] | | | ImageNet-1k | | |
| Early Stop | 0% | 9% | 17% | Val | V2 | ReaL | 0% | 9% | 17% | Val | V2 | ReaL |
|---|---|---|---|---|---|---|---|---|---|---|---|---|
| Val Set (10%) | 81.10 | 78.31 | 75.23 | 73.01 | 60.01 | 79.68 | 89.86 | 85.33 | 82.30 | 71.24 | 58.38 | 78.70 |
| | (1.14) | (1.32) | (0.27) | (0.25) | (0.51) | (0.36) | (0.71) | (1.09) | (1.44) | (0.28) | (0.31) | (0.41) |
| Val Set (1%) | 79.99 | 78.70 | 74.75 | **73.46** | 60.52 | **80.14** | **90.62** | 86.05 | **83.01** | **71.60** | **58.71** | **79.01** |
| | (1.25) | (1.97) | (3.06) | (0.28) | (0.39) | (0.28) | (1.40) | (0.63) | (0.73) | (0.31) | (0.49) | (0.38) |
| LabelWave | 81.00 | 78.37 | 75.02 | 73.02 | 60.05 | 79.66 | 89.84 | 85.14 | 81.15 | 71.23 | 58.36 | 78.70 |
| | (1.28) | (1.23) | (0.27) | (0.08) | (0.28) | (0.34) | (1.12) | (0.90) | (1.89) | (0.37) | (0.46) | (0.40) |
| GD | 79.22 | 77.56 | 74.66 | 67.22 | 54.59 | 74.25 | 89.25 | 84.95 | 81.71 | 71.23 | 58.36 | 78.70 |
| | (11.0) | (1.76) | (1.33) | (13.65) | (13.2) | (12.8) | (1.14) | (0.63) | (1.26) | (0.37) | (0.46) | (0.40) |
| GWA | **81.57** | **78.93** | **75.70** | 73.28 | **60.53** | 79.95 | 89.73 | **86.08** | 82.55 | 71.25 | 58.62 | 78.74 |
| | (0.96) | (0.91) | (0.80) | (0.23) | (0.53) | (0.13) | (0.40) | (2.30) | (0.56) | (1.25) | (1.15) | (1.24) |

Our work proposes using GWA as a trustworthy validation metric during optimization. To this end, we evaluate if GWA provides a reliable signal to monitor training progress and determine when training should be stopped. Concretely, we use *the optimization step during which the maximum GWA value is reached* (after a warm-up period of 10% of the total training steps) as an early stopping point. We conduct a large-scale empirical evaluation of various early stopping criteria across diverse datasets and architectures. We train ConvNeXt and ViT/S-16 models from scratch on CIFAR-10, CIFAR-10-N (with varying levels of label noise), and ImageNet. For each dataset, we compare the final test accuracy achieved when employing different early stopping strategies: (1) traditional train/validation splits (90/10% and 99/1%), (2) LabelWave (measuring prediction change per sample) and Gradient Disparity (GD, average pairwise gradient $\ell_2$-distance), and (3) our proposed GWA. GWA is evaluated in a fully supervised setting utilizing the entire training dataset with an additional ablation on train set size in Tab. 5.

Results in Tab. 1 indicate that GWA matches or outperforms most other metrics across datasets and model architectures when used as an early stopping criterion. Specifically, on CIFAR-10/CIFAR-10-N, GWA achieves an on average 0.4% higher test accuracy compared to standard GWA validation splits, and 0.67% over LabelWave. In [28] two early stopping criteria are proposed for GD: the fifth inter-epoch increase, or an increase for 5 consecutive epochs. Both fail completely in our case, with the former criteria stopping consistently too early and the latter criteria not being triggered in most of our experiments (*see also* Fig. 2). This is also the reason why the test accuracies of LabelWave and GD are identical for ConvNeXt on ImageNet-1k, as both did not stop early. While the 1% validation set slightly outperforms GWA on the ConvNeXt, GWA beats the 10% baseline and provides strong results when dealing with input/label noise (CIFAR-10 [9%, 17%], V2). On ViT, GWA even outperforms the 99/1% validation split strategy commonly used in literature while eliminating its reliance on held-out data. This performance is particularly strong on the smaller CIFAR-10 dataset while also scaling to ImageNet.

**Alignment Distribution**    To gain a deeper understanding of the information captured by GWA, we track its evolution through training compared to validation accuracy and LabelWave's *prediction change* on identical experimental runs. The **left** panel of Fig. 2 demonstrates that both GWA and LabelWave closely track validation accuracy during training with little to no label noise (CIFAR-10), with convergence setting in towards the end. However, the **center** plot reveals GWA's superior sensitivity to early symptoms of overfitting due to the presence of label noise within CIFAR-10-N. This heightened sensitivity results in choosing an early stopping point very close to the one determined by validation accuracy, despite not requiring a validation set in the first place. On the other hand, relying on LabelWave in this label noise setting is not helpful, as overfitting is not detected. Since LabelWave did not outperform the validation set baselines in any of our experiments, we exclude it from the further evaluations below.

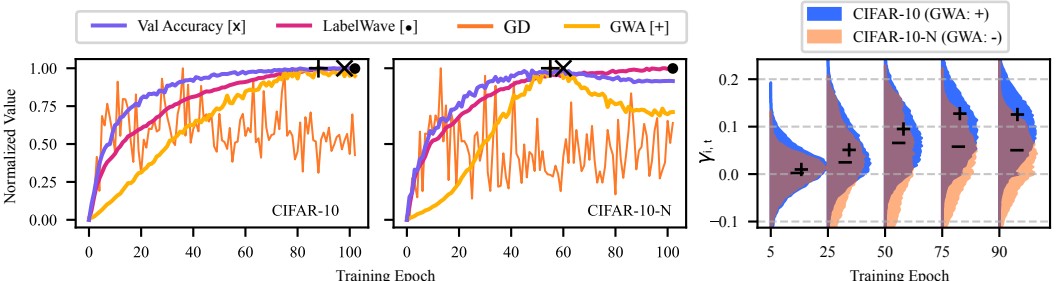

Figure 2: GWA tracks validation accuracy and captures subtle training dynamics associated with generalization (**left, center**) better than LabelWave. Line plots depict normalized values of validation accuracy (10%), LabelWave's *prediction change* and GWA's corrected mean across training. Markers indicate time step for early stopping according to each criterion. The underlying distribution of alignment scores $\gamma(x_i, \mathbf{w}_T)$ (**right**) at time $T$ can be seen as a cross-section providing further insights into training. Label noise highly influences properties of the CIFAR-10-N distribution vs. CIFAR-10.

Focusing on the distributional nature of GWA, the **right** panel in Fig. 2 illustrates the evolution of the underlying distribution of alignment scores across training. Both CIFAR-10 and CIFAR-10-N exhibit unimodal distributions approximating Gaussians to different degrees throughout training. However, there are notable difference between the two dataset variants. The "clean" CIFAR-10 dataset displays a more concentrated distribution with higher GWA values (**right**, indicated by **+/−**) throughout training. Conversely, CIFAR-10-N exhibits consistently lower GWA values, with the substantially larger proportion of negatively aligned samples reflecting the impact of noisy labels on gradient coherence. This analysis confirms that GWA not only captures temporal dynamics but that properties of the underlying per-sample alignment distribution also provide valuable insights into data quality, offering a richer understanding of training behavior than solely considering validation accuracy.

## 4.2 Model Comparison

Next, we investigate whether GWA remains consistent across multiple model initializations, dataset variants and model architectures. We require such consistency to be able to reproducibly use GWA in practice, for example by monitoring it across hyperparameter sweeps. For this purpose, we analyze the correlation between the maximum alignment achieved during training ($\max_T \mathbb{E}[\mathcal{A}_T]$) on CIFAR-10 and its label noise variants (CIFAR-10-N) and the final test accuracy, using both ConvNeXt and ViT models. The resulting scatter plot (Fig. 3 **left**) reveals a strong positive correlation between the test performance of models trained on more or less noisy dataset variants (leading to variations in test accuracy) and GWA. This holds across model families.

To further assess robustness beyond label noise, but also to the effect of domain shifts, we also evaluated test accuracy on CIFAR-C – a version of the standard CIFAR-10 test set incorporating realistic input perturbations such as blurring or image noise (Fig. 3 **center**). Again, a strong correlation is observed between the models' test accuracy and GWA across both model families. These observations are quantitatively supported by the correlation statistics presented in the table (Fig. 3 **right**), which demonstrate high Pearson and Spearman correlation coefficients. These results highlight GWA's capacity to provide a consistent measure enabling reliable model quality comparisons.

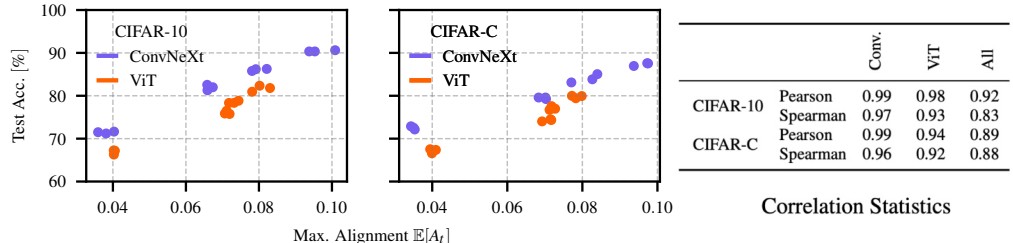

Figure 3: Maximum alignment $\mathbb{E}[\mathcal{A}_T]$ allows for comparing model performance across runs. Scatter plot (**left**) shows correlation between $\mathbb{E}[\mathcal{A}_T]$ and test accuracy on CIFAR-10 and with varying performance on its label noise variants for ConvNeXt and ViT. Correlation is even stronger when evaluating against robustness benchmark CIFAR-C (**center**). Pearson and Spearman correlation coefficients for all cases (**right**) corroborate visual findings ($p < 0.001$).

**Model Robustness**    As standard test sets are often from the same domain or even identical initial parent dataset as the training and validation set, we additionally evaluate models selected using the early stopping criteria studied above on CIFAR-C and ImageNet-C (Tab. 2), two established benchmarks that apply a diverse range of image corruptions to standard test sets. This allows us to evaluate if GWA actually detects training dynamics that extend out of strict in-domain learning and detect models that work on corrupted data that mimics real-world perturbations. Our results reveal that models chosen via GWA-based early stopping consistently exhibit improved performance across various corruption types compared to those selected using traditional validation accuracy. Specifically, we observed an average improvement of $0.55\%$ on CIFAR-C and $0.67\%$ on ImageNet-C compared to models trained with a $10\%$ validation set. Our findings show that GWA not only correlates with test performance, enabling early stopping, but also enhances model robustness to real-world input perturbations, effectively closing the loop from addressing label noise to mitigating input noise.

Table 2: Using GWA to determine early stopping results in models that are more robust to realistic perturbations and suitable for deployment. Test accuracy averaged across 3 runs with ViT/S-16.

| | Test Accuracy [%] | | | | | | | |
|---|---|---|---|---|---|---|---|---|
| | CIFAR-C | | | | ImageNet-C | | | |
| Model | Blur | Digital | Noise | Weather | Blur | Digital | Noise | Weather |
| Val Set (10%) | 81.19 | 79.42 | 77.08 | 79.25 | 55.78 | 64.23 | 62.43 | 60.06 |
| Val Set (1%) | -0.88 | -1.09 | -0.68 | -1.04 | **+0.59** | +0.44 | +0.43 | +0.57 |
| GWA | **+0.52** | **+0.53** | **+0.60** | **+0.56** | +0.57 | **+0.61** | **+0.93** | **+0.60** |

## 4.3    Connecting Model Performance to Training Data

Our proposed GWA is inherently linked to the training data itself through the per-sample alignment scores constituting the distribution. These per-sample alignment scores offer additional insight into the training process which traditional validation metrics lack. To demonstrate the value of this distributional quantity, we examined individual per-sample alignment scores during optimization, and the corresponding characteristics of the training images. We established in Sec. 3 that negative alignment scores likely correspond to outliers, rare examples, or mislabeled samples opposing the overall "training direction", while positive values indicate that samples are aligned with the currently learned patterns, *i.e.*, representing "general" features/concepts early in training and more specific yet common features/concepts later on; this mirrors the simplicity bias argument of [14] and others.

Figure 4 showcases example images from CIFAR-10 and its noisy variant (CIFAR-10-N) with highest and lowest per-sample alignment at epochs 5, 50, and 90 for the *dog* and *car* classes. A striking observation is that nearly all samples with negative alignment scores when training on CIFAR-10-N are mislabeled – a major benefit of using GWA achieved without employing an explicit outlier or mislabelling detection technique. Furthermore, analysis of CIFAR-10 reveals that samples with high

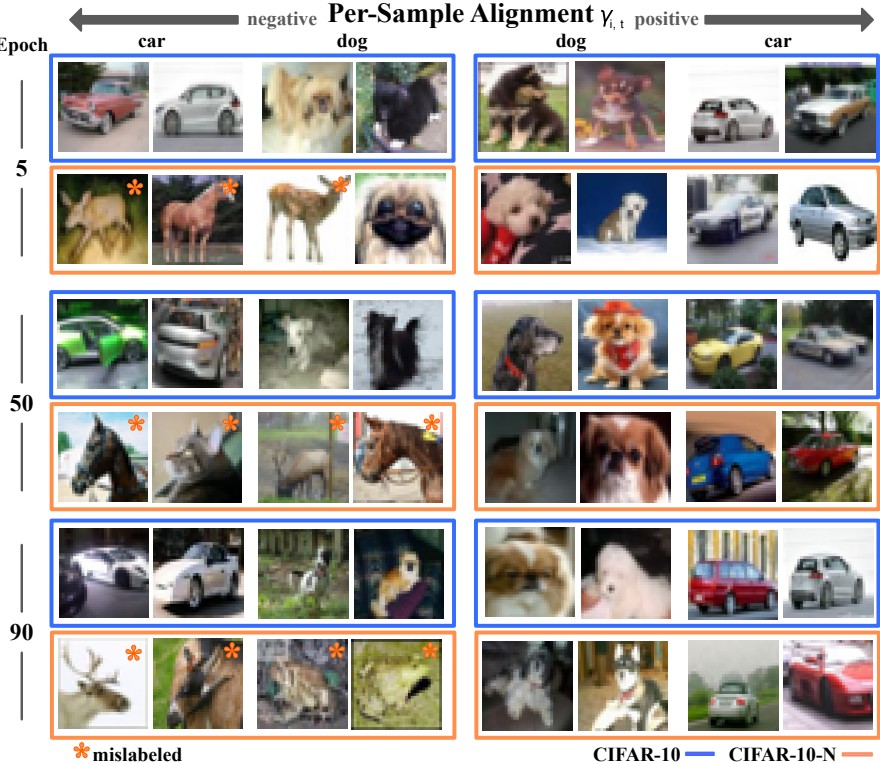

Figure 4: Per-sample alignment scores $\gamma(x_i, \mathbf{w}_T)$ reveal insights into data characteristics and learning progression. Example images from CIFAR-10 and CIFAR-10-N with highest and lowest alignment scores at epochs 5, 50, and 90 of training. Images displayed for the *dog* and *car* classes.

positive alignment scores tend to be visually simpler, while those with negative alignments are more cluttered and/or visually challenging. Notably, positively aligned samples in the beginning at epoch 5 predominantly feature easily classifiable instances (*e.g.*, frontal views of cars with white background, dog faces), whereas later epochs showcase increasingly complex yet still representative examples (rear view of cars, dog faces with long ears). This pattern corroborates the aforementioned prior work stating that models initially learn from simpler samples before progressing to more complex features – a dynamic directly reflected in the per-sample alignment scores.

## 4.4 Fine-Tuning

Our previous analyses focused on training models from randomly initialized weights, representing a scenario where the model learns all relevant information during the optimization process. However, modern deep learning frequently leverages pre-trained models via fine-tuning – fundamentally altering the initial conditions and subsequent training dynamics. We next investigate how this impacts our proposed GWA metric. As visualized in Fig. 5, GWA exhibits significantly higher values after the first epoch of fine-tuning, reflecting the immediate benefit of pre-trained features for generalization – corroborated by high initial accuracy. Unlike training from scratch where GWA typically consistently increases, we observe an initial

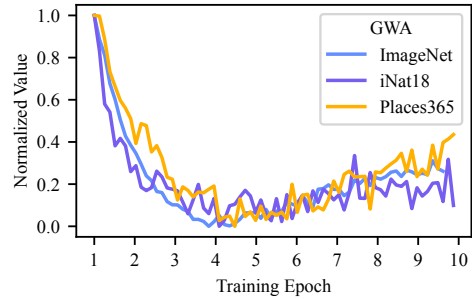

Figure 5: Initial GWA decrease during fine-tuning before increasing equivalent to training from scratch. ImageNet-21k pre-trained ViT/B-16 fine-tuned for 10 epochs on ImageNet-1k /Places365, 20 epochs on iNat18.

dip during the early stages of fine-tuning. This suggests the model must first adapt to dataset-specific details, temporarily disrupting the initially strong alignment. After a few epochs, this trend reverses, mirroring the increasing alignment observed during training from scratch. Consequently, when employing GWA for early stopping during fine-tuning, we now prioritize identifying the initial minimum in alignment before taking the maximum GWA to determine early stopping. As shown in Tab. 3, this refined approach yields reliable performance, with test accuracy outperforming the validation-based metrics on most datasets.

## 5    Conclusion

We introduced GWA and investigated its ability to capture dataset- and sample-level training dynamics during optimization, as well as serve as a robust early-stopping criterion. We have shown that GWA is a reliable metric that matches or surpasses classical validation sets on a range of tasks, while offering unique insights into training by connecting performance directly to individual training samples – a capability particularly valuable in the presence of noise and low-data regimes. In future work, we intend to expand our evaluation beyond supervised image classification, *e.g.*, to the self-supervised setting, where the importance of gradient directions has recently been noted [46], and to other modalities such as text, where

Table 3: GWA matches or outperforms other early stopping criteria when fine-tuning a ViT/B-16 pre-trained on ImageNet-21k. Top-1 test accuracy averaged across 3 seeds, min-max range below in gray.

| | Test Accuracy [%] ($\triangle$ min–max) | | | | |
|---|---|---|---|---|---|
| | ImageNet-1k | | | iNat18 | Places365 |
| Early Stop | Val | V2 | ReaL | | |
| Val Set (10%) | 84.04 | 73.94 | 88.96 | 72.87 | 58.66 |
| | (0.07) | (0.35) | (0.05) | (0.07) | (0.03) |
| Val Set (1%) | 84.11 | 74.19 | 89.00 | 73.65 | **58.86** |
| | (0.07) | (0.06) | (0.06) | (0.37) | (0.35) |
| GWA | **84.15** | **74.32** | **89.05** | **73.73** | 58.78 |
| | (0.06) | (0.26) | (0.03) | (0.14) | (0.29) |

applications to the autoregressive loss are a natural evolution of our results. Moreover, seeing as current techniques are either too inefficient [1] or ineffective [23] for large-scale applications, we hope that our work sparks interest in computationally efficient train-time generalization proxies such as GWA.

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

# Appendix

Here, we provide further evaluation of our scalable estimator (Appendix A.1), compare against related work not directly usable as a train-time generalization proxy, yet relevant to our approach (Appendix B), and provide more details on our empirical evaluation as well as the results (Appendix C).

# A Scalable Estimator

## A.1 Estimator Bias

The proposed efficient online estimator of GWA is computed continuously during each epoch. This estimator's key difference from a pure *offline estimator* is that the weights $w$ are not fixed; they drift with each mini-batch update. This drift is the source of a systematic bias, and its characterization depends critically on the point of comparison.

**Online Bias Relative to Fixed Time Point (constant $w$)**  Compared to the GWA value at the epoch's starting weights $w_0$, the online estimator exhibits a bias directly governed by the learning rate $\eta$. This can be shown by using a first-order Taylor expansion on the expected alignment $A(w)$, where the bias for a measurement at step $t$ is determined by the change in $w$, approximately given by $\nabla A(w_0)^\top (w_t - w_0)$. Since the weight displacement $w_t - w_0$ is the result of accumulated gradient steps, each scaled by $\eta$, the average bias over the epoch is linearly proportional to the learning rate. A larger $\eta$ causes greater drift from $w_0$, resulting in a larger first-order bias. In practice, however, this bias is negligible as seen in Fig. 6: both the online and offline estimator have near-perfect correlation. Quantifying the bias between online and offline estimators (at the end of each epoch) shows a small effect size [47] for $\eta = 0.1$ and SGD with a mean of $0.04$ (maximum $0.12$), decreasing further for $\eta = 0.01$ to $0.027$ (maximum $0.08$), and to $0.020$ (maximum $0.05$) for $\eta = 0.001$.

In addition, the existing bias is better understood not as an error, but as a temporal shift. If we re-frame the comparison against the metric at the epoch's midpoint $w_{\mathrm{mid}}$, the dominant first-order bias cancels out. This is because the weight updates $w_t$ are, to a first approximation, distributed symmetrically around $w_{\mathrm{mid}}$, causing the average displacement $\mathbb{E}[w_t - w_{\mathrm{mid}}]$ to be near zero. However, the GWA metric defined in the paper (Eq. (2)) is not just the mean alignment but a ratio involving the distribution's *kurtosis*. Analyzing higher-order moments like kurtosis requires a second-order Taylor approximation. However, while the online GWA is not perfectly unbiased with respect to the midpoint, its bias is of a higher order.

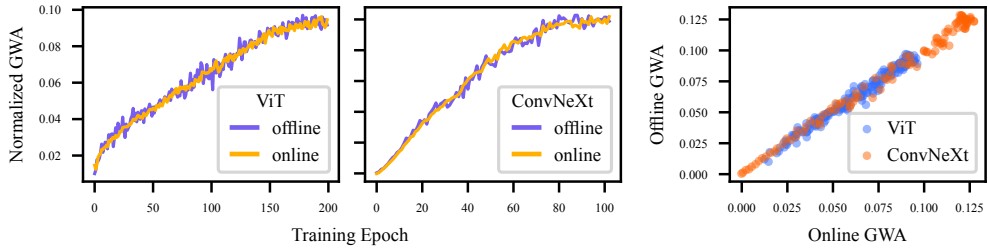

Figure 6: GWA computed online during the forward pass using the efficient scalable estimator detailed in Algorithm 1 nearly exactly matches the offline (after each epoch) computation of GWA for both ViT and ConvNeXt trained on CIFAR-10 (*left, center*) with near-perfect correlation across the whole training (*right*).

**Online Per-Epoch vs. Step-Wise Offline Estimators**  An alternative, mini-batch perspective on the estimator's bias is through the lens of a hypothetical *step-wise offline estimator*, which would perform a full (and computationally prohibitive) offline evaluation on the entire dataset at each actual weight vector $w_t$ (*i.e.*, after every single update step). Instead of computing the metric over the full dataset at each time step, our estimator only computes the alignment over a smaller set $\mathcal{G}_t$, equal to the batch size $b$. In this case, the bias of the estimator would thus be equivalent to the bias induced by taking the batch as a representation of the full dataset. For smaller batch sizes, however, this

can lead to instabilities when computing the corresponding moments of $\mathcal{A}_t$ for GWA. Our proposed online estimator for GWA can be seen as a special, more efficient case of this step-wise mini-batch estimator that does not have these instability problems. Instead of computing the moments for $\mathcal{A}_t$ based on a single batch, we take the alignment scores $\gamma(x_i, w_{t(i)})$ across the entire epoch, but with changing weights $w_t$, to compute $\mathcal{A}_t$. This "epoch average" effect inherently results in a smoothed version of the step-wise offline estimator, as seen in Fig. 6. It sacrifices temporal accuracy at each step for a computationally efficient, stable, and interpretable summary of the epoch's overall alignment dynamic. For settings with larger batch sizes, and potentially also no clear definition of an epoch such as in NLP, computing $\mathcal{A}_t$ on a batch-level can be a reasonable approach, closing the gap to the step-wise offline estimator.

To summarize, while the proposed online estimator is neither unbiased nor easily allows for quantifying its bias (see Sec. 3 and discussion above), we find that its bias, depending on the properties of the batch and the change in weights across the interval it is measured on, are neglectable in our evaluations. The scalable GWA estimator thus allows for (1) a minimal train-time and implementation overhead, and is (2) near-perfectly correlated with the offline estimation of GWA in practice.

## A.2 Full Model vs. Linear Classifier

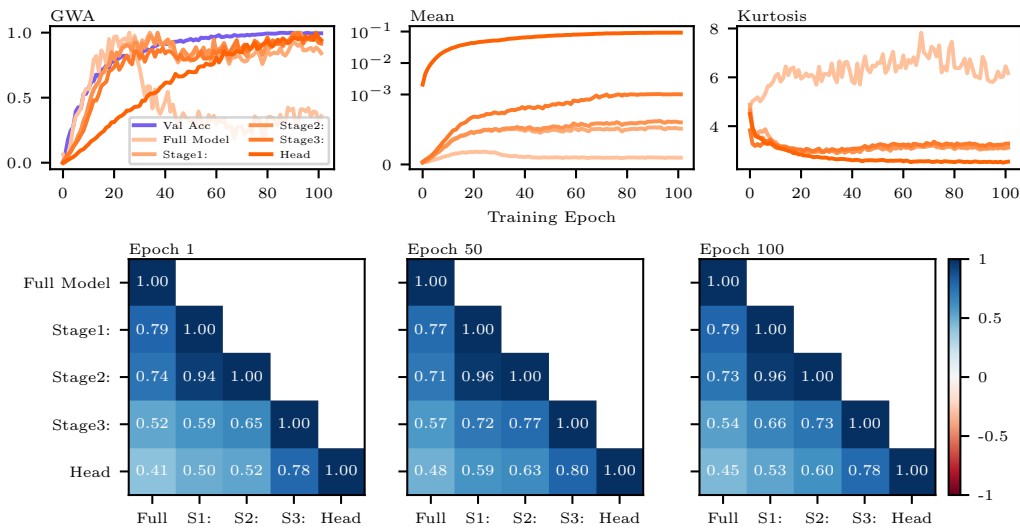

Figure 7: Computing alignment scores using the linear classifier head mitigates dimensionality issues such as decreasing mean of the alignment distribution and increasing tailedness (*top center and right*). Using the full model, or including more layers of the network in addition to the head layer (*e.g.*, from the second residual block onwards in a ConvNeXt, denoted as "Stage2:"), leads to a worse learning signal (*top left*). This is corroborated by a decrease in correlation between these scores and the alignment scores computed with the classifier head (*bottom*).

To mitigate problems caused by the high dimensionality of modern networks (*e.g.*, noise, computational cost), our approach focuses on the linear classifier head with closed-form per-sample gradient computation as introduced in Sec. 3. The plots in Fig. 7 show that while including more layers still reflects certain patterns during optimization, such as an initial increase in alignment. While this initial increase in dataset alignment is visible throughout the model, the earlier layers quickly lead to an obfuscation of the signal. Gradient updates become increasingly more orthogonal when looking at the full model (*top center*) while also having relatively more updates near the bounds $[-1, 1]$ of our alignment range (*top right*).

While leveraging the linear classifier head provides a stronger learning signal, alignment score magnitude potentially remains sensitive to latent representation size – possibly hindering cross-architectural comparison due to cosine similarity degradation in high dimensions. We address this using a JLT to reduce dimensionality while preserving pairwise distances. Fig. 8 demonstrates that JLT mitigates this issue, increasing expected alignment scores $\mathbb{E}[\mathcal{A}_T]$ for models with larger latent spaces, and allowing for more comparable GWA values across architectures despite inherent

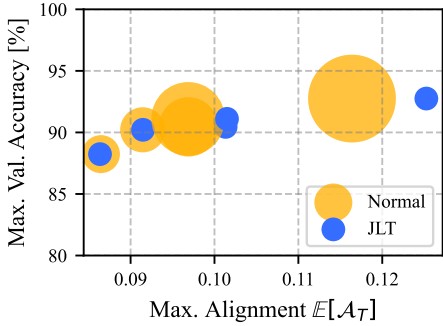

Figure 8: Alignment scores for ConvNeXt models with latent embeddings of increasing size (indicated by circle size) and after applying the Johnson-Lindenstrauss Transform (JLT) to reduce embedding dimensions to a constant size of 192.

differences in model design. Notably, even without applying the JLT, the alignment scores correlate well with validation accuracy across embedding sizes.

## A.3 Kurtosis Correction

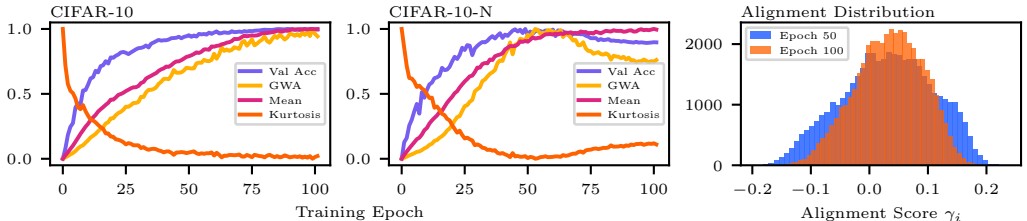

Figure 9: Reproduction of Fig. 1 with mean and kurtosis components of GWA plotted individually. In simple cases, when the ConvNeXt learns without problems (*left*) the mean can be a sufficient proxy for generalization. Often, more sophisticated patterns can, however, only be detected by looking not only at the distributions mean but also it other properties. Notably, kurtosis seems to be a good correction factor to account for changes in the distributions shape (*center, right*).

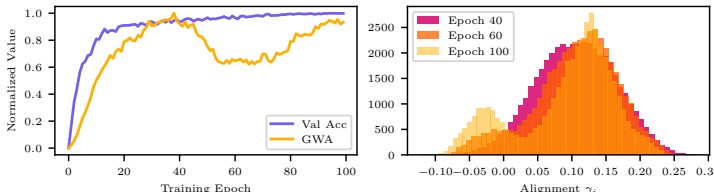

Figure 10: While the alignment distribution tends to be unimodal in our evaluations, there is no guarantee for such behavior. In some cases, analyzing the alignment distribution directly can help to better understand training dynamics and dataset characteristics.

GWA is based on a distribution of values for each sample in the dataset. This allows us to connect training performance directly to individual samples, but it also introduces a challenge: we need a single, representative number to track how this entire distribution changes over time. Describing a changing distribution with just one number is difficult, especially when its shape can be complex. We propose to quantify the mean shift of the alignment distribution together with a kurtosis correction to create a robust summary statistic. This correction is essential because the distribution of alignment scores can not be assumed to be Gaussian. Consistent with the long- and heavy-tail theories of deep learning, we find the kurtosis is suitable correction to account for changes in the tails of the

distribution, which we find to be particularly important for noisy and challenging datasets like CIFAR-10-N (*see* Fig. 9, *center and right*) or ImageNet.

While our corrected GWA metric works well in most cases, it has limitations. These failure cases can often be spotted by looking at the plot of the underlying alignment distribution at specific epochs (*e.g.*, Fig. 10). One concrete issue is the occurance of the distribution becoming bimodal, which, if it happens, tends to be late during training. Investigating the learning dynamics that cause these specific distributional shifts is a promising direction for future research.

### A.4 Interpolation of Noise

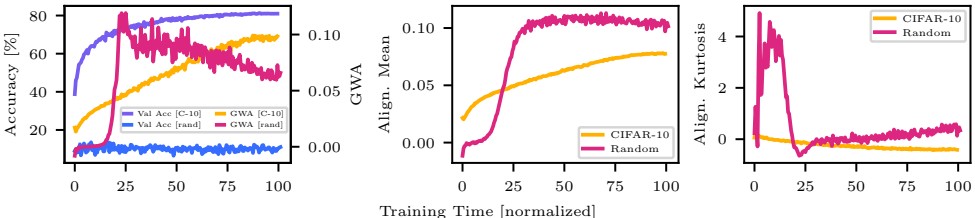

Figure 11: Training with random labels forces the model to learn only sample-specific information (*label memorization*) and random guessing accuracy compared to actual generalization. GWA, capturing all training dynamics, reflects this non-generalization. While alignment does increase with random labels, the mean and kurtosis show a distinct pattern, different to standard smooth generalization. The initial low mean alignment around 0 with corresponding high kurtosis across the dataset flips completely when the network starts to memorize.

To fully understand GWA as a generalization proxy, we train models on CIFAR-10 with completely randomized labels, following the setup in [48]. While this scenario of training on pure noise is unrealistic for any practical application, it is an informative method to isolate label memorization and evaluate GWA's response. In this setting, the model cannot learn generalizable patterns and is forced to memorize the labels, leading to validation performance equivalent to random guessing. As shown in Figure 11, GWA exhibits a distinct pattern under these conditions. Initially, the mean alignment is near zero with high kurtosis. As the network begins to memorize the random labels, GWA increases sharply, driven by a rapid rise in mean alignment accompanied by a substantial reduction in kurtosis. After this peak, GWA begins to decrease even as the mean alignment stays high. Thus, rather than remaining static near zero as might be expected, GWA characterizes memorization through a distinct dynamic pattern: a sharp initial rise followed by a decay. This reveals how the metric captures the distinct phases of the model memorizing a noisy dataset.

## B  Related Work

Next, we provide a comparative analysis of GWA to previously introduced techniques for generalization gap quantification and training dynamic assessment (summarized in Sec. 2). This highlights the benefits of our efficient train-time proxy for generalization and early stopping. Note that the focus of GWA is not exactly the same as the other metrics shown here, but we believe the comparison to be useful nonetheless.

### B.1 Gradient Norm and Second Order Information

Gradient norm is a common metric used for tasks such as approximating second-order information to analyze loss landscapes and sample influence (*see e.g.*, TracIn [7]). Fig. 12 shows, however, that gradient norm is not a replacement for GWA to validate generalization. While both the ConvNeXt and ViT models generalize, the average gradient norm develops differently for both architectures (*right*). While we observe weak overall correlation between the per-sample gradient norms and our alignment scores $\gamma_i$ (as defined in Algorithm 1), we see discernible patterns: a reduction in gradient norm towards the end of training tends to correlate with an increase in alignment, while higher relative gradient norms — particularly at the end of training — correlate with alignment scores indicating

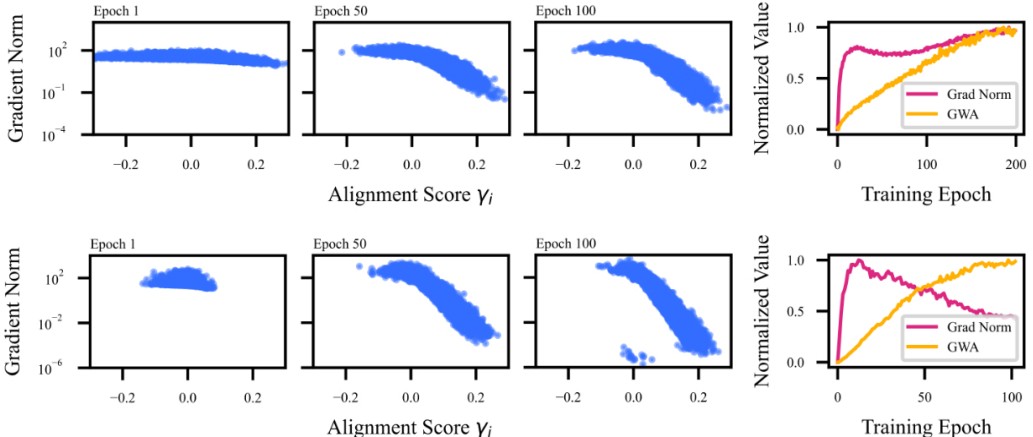

Figure 12: Per-sample gradient norm and alignment scores $\gamma_i$ for ViT (*top*) and ConvNeXt (*bottom*) trained on CIFAR-10 show that both measures quantify distinct characteristics of the training dynamics. GWA is consistent across both architectures and better reflects generalization during training (*right*).

orthogonal gradient updates. An interesting exception is observed in our ConvNeXt experiment (*bottom*), where a group of samples exhibits the smallest gradient norm and low alignment at the end of training, suggesting these examples are well-learned with remaining loss reduction being sample-specific, *i.e.*, inherently orthogonal to general optimization.

Notably, unlike GWA, metrics such as TracIn are susceptible to the unbounded property of gradient norms and thus sensitive to their volatility and outliers. In the right-bottom panel of Fig. 12 the gradient-norm distribution is heavily skewed, with a peak of $7.16 \times 10^3$ at epoch 12. By contrast, GWA's distribution is compact, only depending on gradient direction and therefore unaffected by magnitude outliers.

Spearman analysis on CIFAR-10-N from Fig. 2 when overfitting occurs shows that GWA correlates strongly with validation accuracy (R= 0.97) yet poorly with training accuracy (R= 0.56) during overfitting, as desired. In contrast, the gradient norm exhibits a high correlation with training accuracy (R= 0.90) but a weak one with validation accuracy (R= 0.24). This also holds when only considering the classifier head, similar to GWA, where both gradient norm and also the second-order Hessian trace reach perfect correlation with training accuracy (R= 1.0) yet moderate correlation with validation accuracy (R≈ 0.48).

In summary, while per-sample gradient norms do not directly correlate with GWA and thus cannot be used as a generalization proxy, analyzing both helps understand which samples have been learned and which remain hard to learn.

## B.2 Connection to Loss Landscape Geometry

Limited empirical work has analyzed the alignment between gradients and model weights, with [32] being a notable exception; they analyze cosine similarity between the mini-batch estimate of the gradient $\mathbf{g}$ and the difference between the current weights $\mathbf{w}_T$ and "optimal" weights $\mathbf{w}^*$, defined as the final weights of a *previous run with the same seed*. With this metric they aim to quantify the geometric properties of sampled gradients along optimization paths within the loss landscape. Fig. 13 shows that GWA and $\mathbf{w}^*$-alignment share similar patterns, converging towards comparable behavior in the later parts of training. However, $\mathbf{w}^*$-alignment *requires two full training runs*, making it computationally expensive or even infeasible for very large model training. GWA offers a compelling alternative: capturing gradient alignment from a different perspective yet with similar insights, significantly reduced computational cost, and potentially broader applicability. We regard a combination of the efficiency, reliability and validity of GWA with the theoretical insights of [32] a promising future research direction.

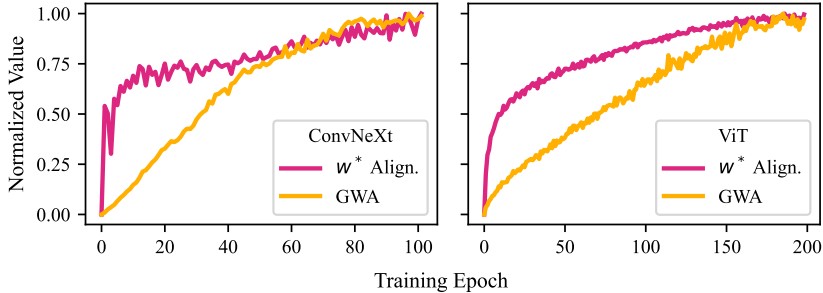

Figure 13: GWA mirrors the alignment between mini-batch gradients and the direction to the optimal weights $\mathbf{w}^*$ towards the end of training but without requiring the expensive computation of $\mathbf{w}^*$.

### B.3 Pairwise Gradient Alignment

Relatedly, prior research (discussed in Sec. 2) analyzes gradient direction via pairwise per-sample gradient alignment proposing metrics such as *gradient coherence* and *stiffness* [1, 25, 26, 27]. Most approaches in this area are related and measure very similar quantities, a proxy for which is the pairwise per-sample gradient alignment (*i.e.*, the pairwise gradient cosine similarity). Note that none of these quantities actually use the weights, contrary to GWA. Fig. 14 reveals that the average pairwise gradient alignment exhibits high variance - especially early on and likely due to initial training randomness - while GWA is more stable. After the initial differences, the two measures exhibit similar trends in mid-to-late in training. However, GWA exhibits fewer fluctuations in this mid-to-late stage, providing a much more consistent signal. Moreover, the substantial computational overhead of pairwise per-sample gradient alignment – requiring full model gradients across multiple timesteps, their storage, and computationally expensive pairwise cosine similarity calculations – limits its applicability.

In summary, compared to prior works, GWA offers a compelling complementary approach. It provides comparable insights into gradient alignment but has significantly improved efficiency and thus unlocks new possibilities for large-scale research in this domain. This renders GWA an attractive tool to re-evaluate existing work and accelerate progress in understanding neural network training dynamics.

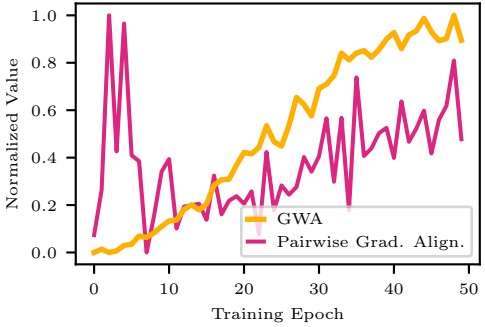

Figure 14: GWA exhibits substantially more stable relative alignment during the entire training duration compared to pairwise per-sample gradient alignment.

### B.4 Neural Collapse

Neural collapse [49] is an important theoretical phenomenon which describes the terminal-phase collapse of last-layer features to their structured class-means. GWA offers a complementary view based on the per-sample gradient-weight interaction throughout training on realistic, noisy data, where quantifying variance is the key signal for generalization. Recent work on neural collapse has introduced matrix information theory to analyze training dynamics. [50] use the latent embeddings of

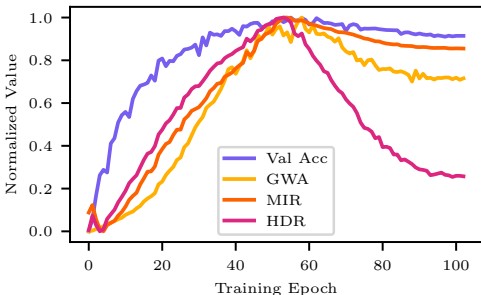

Figure 15: GWA exhibits substantially more stable relative alignment during the entire training duration compared to pairwise per-sample gradient alignment.

each samples together with class-dependent classifier weights to compute matrix mutual information based on Gram matrices. Fig. 15 shows that the two values introduced in [50] based on matrix information theory, matrix mutual information ratio (MIR) and matrix entropy difference ratio (HDR), correlate well with GWA. We believe further connecting these two approaches in future work is not only a promising way to integrate GWA and information-theoretic research but also allows for more efficient information theoretic approaches, without full Gram matrices, and providing GWA's per-sample insights.

## C    Further Details on the Experimental Setup

### C.1    Implementation Details

Table 4: Training hyperparameters for respective model-dataset pairs. Setting for training ImageNet-1k from scratch taken from [51]. ImageNet-22k pre-trained weights for fine-tuning and settings are from [44] with adaptations for datasets ImageNet-1k, iNat18, and Places365 in parentheses.

| | CIFAR-10 / CIFAR-N | | ImageNet-1k | | Fine-Tuning |
|---|---|---|---|---|---|
| **Hyperparameter** | ConvNeXt-P | ViT/CIFAR-4 [52] | ConvNeXt-F | ViT/S-16 | ViT/B-16 |
| Optimizer | Adam | Adam | AdamW | AdamW | SGD |
| LR | 0.001 | 0.0001 | 0.001 | 0.001 | (0.01, 0.05, 0.01) |
| Weight Decay | – | – | 0.0001 | 0.0001 | – |
| Momentum | [0.9,0.999] | [0.9,0.999] | [0.9,0.999] | [0.9,0.999] | 0.9 |
| Batch Size | 512 | 256 | 1024 | 1024 | 512 |
| Iterations | 9000 | 35000 | 112650 | 112650 | (26000, 17500, 37500) |
| Image Size | 32 | 32 | 224 | 224 | 224 |
| RandAug [53] | 1 | 1 | 10 | 10 | (0, 2, 0) |
| Mixup [54] | 0.0 | 0.0 | 0.2 | 0.2 | (0.0, 0.1, 0.0) |
| Grad Clip Norm | – | – | 1.0 | 1.0 | 1.0 |
| Scheduler | Cosine | Cosine | WarmupCosine | WarmupCosine | WarmupCosine |
| Warmup Steps | – | – | 10000 | 10000 | 500 |

We provide the training settings for all ConvNeXt models on CIFAR-10 and its variants as well as for ImageNet in Tab. 4. The settings are used for our main results in Tab. 1, Fig. 2, and Fig. 4 (Sec. 4). Tab. 4 also includes the training and fine-tuning settings for all ViT models used in Tab. 1, Tab. 2, and Tab. 3 (*and corresponding Fig. 5*). The ImageNet-22k pre-trained weights of the ViT/B-16 are from [44] and are available here. Results in Fig. 3 for both architectures are obtained with the same settings.

### C.2    Auxiliary Results

**Alignment of Mislabeled Samples**    Beyond early stopping, GWA provides a novel perspective on training dynamics and generalization. Fig. 16 shows the alignment distribution of CIFAR-10-N (40% label noise) across three epochs (*beginning, close-to-optimal, overfitting*). These distributions make up GWA in Fig. 2 (*center*). Initially, most samples have updates that are orthogonal to the initial optimization trajectory, with the alignment distribution being compact, around zero, and with

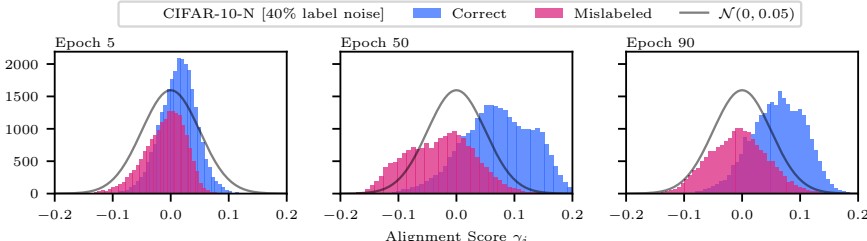

Figure 16: Corresponding alignment distribution of Fig. 2 (*center*) with alignment scores of correctly labeled and mislabeled samples being colored differently. Mislabeled samples tend to be more negatively aligned during training, with the clearest separation between mislabeled and correctly labeled samples around the optimal stopping time step near Epoch 50.

mislabeled examples exhibiting a slightly negative bias. This bias intensifies toward the optimal stopping epoch, differentiating mislabeled and correctly labeled samples. As the model learns, the optimization requires to mislabeled examples to further reduce the overall loss, driving corresponding updates towards zero again – a directional shift from prior learning. At the same time, correctly labeled samples are also pushed towards zero, a potential result of the model overfitting and/or learning samples specific information of these samples. This leads to a concentration of the distribution. Analyzing these distributional changes and proactively modulating this behavior warrants further investigation.

**Results with Fixed Training Set Size** Tab. 1 shows the best-case scenario possible with each approach to fairly demonstrate the performance of all early-stopping criteria. Using a smaller validation set, or no validation set at all, allows for using more samples during training, aiding model performance. With a fixed training set size, a $10\%$ validation set outperforms both a $1\%$ validation set and GWA (Tab. 5). However, despite the artificial restriction of training data size in this ablation, GWA and the $1\%$ validation set remain competitive, indicating that the performance of both approaches is not due to training set size alone.

Table 5: Artificially restricting the available training data leads to larger validation sets with $10\%$ to better approximate generalization behavior compared smaller validation sets ($1\%$) and GWA (*val split of* $90/0$ *in Table*) in comparison to the optimal results achievable with each method in Tab. 1.

|  | Val Split | CIFAR-10 [label noise %] | | | ImageNet |
|---|---|---|---|---|---|
|  |  | 0% | 9% | 17% |  |
| ViT | 90/10 | 81.10 | 78.31 | 75.23 | 73.01 |
|  | 90/1 | -0.07 | -0.01 | -0.20 | -0.08 |
|  | 90/0 | -0.12 | -0.01 | -0.27 | -0.21 |
| ConvNeXt | 90/10 | 89.86 | 85.33 | 82.30 | 71.24 |
|  | 90/1 | -0.01 | -0.02 | -1.12 | -0.18 |
|  | 90/0 | -0.22 | -0.17 | -0.35 | -0.09 |

