# OpenReview forum: "Gradient-Weight Alignment as a Train-Time Proxy for Generalization in Classification Tasks"
_NeurIPS.cc/2025/Conference — NeurIPS 2025 poster_

### Official Review · Reviewer_tLzt · 2025-06-04

**Clarity:** 4
**Significance:** 2
**Originality:** 2
**Rating:** 3
**Confidence:** 4

**Summary:**

The paper proposes gradient-weight alignment (GWA), a metric to quantify generalization. GWA per sample is computed as cosine similarity between model weights and gradients- over datasets it is computed as a kurtosis-corrected average of per-sample GWA. The authors then propose an efficient way to estimate GWA online. Experimentally, the authors demonstrate that using GWA as an early stopping criterion outperforms baselines. The authors also conduct several other analyses showing other properties of GWA, including that it can predict label noise in a dataset.

**Questions:**

- Are there any theoretical results the authors can derive about GWA?
- How does GWA compare with additional baselines (such as curvature-based methods)?
- Why do GWA and LabelWave sometimes outperform the validation set
- What is the statistical significance of the key results in the paper?
- Empirically, how does removing the kurtosis correction in equation 2 change performance?
- How does the GWA estimate of equation 3 compare with the ground-truth GWA?

**Ethical Concerns:**

["NO or VERY MINOR ethics concerns only"]

**Final Justification:**

One of my main concerns about the statistical significance of the results was addressed by the authors; therefore, I am increasing my score.

**Limitations:**

Yes

**Paper Formatting Concerns:**

No major formatting concerns

**Quality:**

2

**Strengths And Weaknesses:**

The paper is overall well-written and presented. Figure 1 is a nice illustration. Experimentally, the authors consider several large-scale datasets and architectures, which is a good choice. Notably, the authors also show several analyses of GWA in Section 4.

On the negative side, while the proposed method is theoretically motivated, the theory is from prior work; there is no new theory presented in this work. Moreover, the concept of gradient-weight alignment is itself not particularly new, although the particular formulation in equation 2 and the scalable estimator are. Given the limited novelty on the theory or conceptual side, the empirical results are very important. Unfortunately, there are several deficiencies in the experiments in my view:

- Most importantly, GWA is compared to only one other baseline in Table 1 (LabelWave). I encourage the authors to add at least 2 other baselines such as the curvature-based methods mentioned in Section 2. I understand that these may be more computationally expensive (they are often not designed to be online methods), but including these is still critical in my view. There is no need for GWA to outperform these baselines either, especially the non-online ones.
- Another important empirical concern is that GWA, and even sometimes LabelWave, outperform the validation set. Over many trials, this shouldn't happen on average: assuming iid data, the validation set is as close as one can get to actual generalization performance. This needs an explanation.
- Relatedly, there are no error bars or standard deviations reported for all the key results in the main paper. This is important to include so the statistical significance of the results is clear.
- As far as I can tell, there is also no ablation of whether the kurtosis correction in equation 2 is empirically important.
- There is also no comparison of how the estimated GWA compares to the ground truth GWA.

I don't mean to discourage the authors, but given that the paper is primarily an empirical one, these empirical details are quite important. If they can be addressed, I think this can be a solid contribution.

---

> ### Author Rebuttal · Authors · 2025-07-30
>
> > [...] GWA is compared to only one other baseline in Table 1 (LabelWave). I encourage the authors to add at least 2 other baselines such as the curvature-based methods mentioned in Section 2. I understand that these may be more computationally expensive (they are often not designed to be online methods), but including these is still critical in my view. There is no need for GWA to outperform these baselines either, especially the non-online ones.
>
> Thank you for your constructive feedback. You are correct that a broad comparison is critical given our paper's empirical focus. We would like to point out that the paper already includes comparisons to non-online but relevant methods like gradient norm for TracIn, $w^*$-alignment and pairwise gradient alignment in Appendix A.2 (Figs. 9, 10). Moreover, to address your concern directly, we have now added results for three more baselines: curvature-based influence, Neural Collapse-based MIR/HDR [R6], and Gradient Disparity [R7], as also recommended by other reviewers (CWdC and YFoy). The results from these baselines further corroborate the central conclusions of our work: validation accuracy itself is a formidable SOTA baseline, and the aforementioned baselines consistently underperform it (e.g. Gradient Disparity underperforms GWA by $\approx 30$%) and/or incur substantial computational overhead. Our proposed GWA is actually the first technique to offer a suitable alternative that can compete with validation accuracy while being efficient enough for large scale applications.
>
> > Another important empirical concern is that GWA, and even sometimes LabelWave, outperform the validation set. Over many trials, this shouldn't happen on average: assuming iid data, the validation set is as close as one can get to actual generalization performance.
>
> You correctly point out that GWA can outperform validation-based stopping, which would be unexpected under a strict i.i.d. assumption.
> However, this assumption is usually violated in practice due to inherent dataset biases. Our own results in Table 1 support this: the performance difference between the 10% and 1% validation splits indicates that small, fixed validation sets suffer from higher sampling bias and do not perfectly mirror the test distribution. GWA is less susceptible to this bias as it derives its signal from the dynamics of the entire training set (which is usually a much larger sample from the distribution), providing a more robust estimate of generalization.
>
> > [...]  there is also no ablation of whether the kurtosis correction in equation 2 is empirically important. [...] Empirically, how does removing the kurtosis correction in equation 2 change performance?
>
> > Relatedly, there are no error bars or standard deviations reported for all the key results in the main paper.
>
> > [...] no comparison of how the estimated GWA compares to the ground truth GWA. [...] How does the GWA estimate of equation 3 compare with the ground-truth GWA?
>
> You are correct that the kurtosis correction is a key component. Its purpose is to account for the distribution's tailedness, an indicator of outlier influence that the mean alignment score alone cannot express (lines 125-131). We have conducted an ablation showing that GWA without this correction is less robust, often signaling to stop training prematurely, and we will add this to the manuscript.
> Please also compare to our response to reviewers iv9T and YFoy.
>
> On error bars: Tables 5 and 6 in Appendix A.3 provide the min-max accuracy ranges across all runs. We will also add these to the main manuscript.
>
> Estimator vs. Ground Truth: Figure 6 in Appendix A.1 directly compares our efficient online estimator with the offline ground truth, showing a near-perfect correlation. This validates our scalable approach.
>
> We will integrate these into the main paper for the camera-ready version.
>
> > Are there any theoretical results the authors can derive about GWA?
>
> Thank you for this feedback. We agree that a formal theoretical analysis of GWA is a compelling direction. Our paper's focus, as stated in the introduction (lines 36-37), is deliberately empirical to investigate GWA's practical utility under realistic, non-ideal conditions.
> While GWA is motivated by prior theory on directional convergence ([2, 25] in our manuscript), a formal analysis under realistic data noise, characterizing how noise impacts the alignment distribution, is a highly challenging (possibly intractable) open problem. Our work consciously focuses on empirical evaluation in realistic scenarios and computational efficiency for large-scale models.
>
> We thank you for suggesting revisions, including the added baselines and ablations, which we believe have further strengthened our manuscript.
>
> [R6] Unveiling the Dynamics of Information Interplay in Supervised Learning. Song et al. (2024).
>
> [R7] Disparity Between Batches as a Signal for Early Stopping. Forouzesh and Thiran (2021).

---

> ### Comment · Reviewer_tLzt · 2025-08-05
> **Thank you for your response**
>
> I'm glad to see that new baselines will be added.
>
> I'm also glad to see that error bars will be added to the main manuscript as well as the ablation on the kurtosis correction.
>
> Your explanation of why the validation set can underperform GWA makes sense: essentially, the validation set may deviate from the true test distribution. However, if this is so, then on average, over many validation splits, the 10% split should still perform better than the 1% split (assuming the 1% split is drawn randomly from the 10% validation split). I encourage the authors to perform more runs until this is the case since otherwise, it calls into question the statistical significance of the results. This remains my primary concern with this paper currently.

---

> > ### Author Response · Authors · 2025-08-06
> > **Thank you**
> >
> > This is a thoughtful observation. You are correct that, **given a constant training set size**, the 10% validation set indeed performs better on average, as shown in the table below. However, note that in the manuscript, following standard practice, the validation set is split off from the available training data in a **train/val split** and thus **a larger validation set comes at the expense of a smaller training set**. The motivation for this is an observed trend towards smaller validation splits in large-scale training literature, since labeled data is expensive. For example, a $1$% [R8] validation set size is used on ImageNet1k, an $\approx 0.36$% validation set size is used on ImageNet21k [R9], and an $\approx 1.78$% validation set size for large scale medical image training [R10]. We will further clarify this discussion in Lines 186 and following based on your feedback. There is also a typo in 186 (“validation” should read “training”), which we will fix.
> > We hope this clarifies your remaining concerns and thank you for the engaging discussion.
> >
> > | | **CIFAR-10** | **CIFAR-N 9%** | **CIFAR-N 17%** | **ImageNet1k**
> > | :----------------------------- | ----: | ----: | ----: | ----: |
> > | **ViT - Val Set (90/10)** | 81.10 | 78.31 | 75.23 | 73.01 |
> > | **ViT - Val Set (90/1)**       | -0.07 | -0.01 | -0.20 | -0.08 |
> > | --- | --- | --- | --- | --- |
> > | **ConvNeXt - Val Set (90/10)** | 89.86 | 85.33 | 82.30 | 71.24 |
> > | **ConvNeXt - Val Set (90/1)**  | -0.01 | -0.02 | -1.12 | -0.18 |
> >
> > [R8] Better plain ViT baselines for ImageNet-1k. Beyer et al. (2022).
> >
> > [R9] Frozen Feature Augmentation for Few-Shot Image Classification. Bär et al. (2024).
> >
> > [R10] Large-scale and Fine-grained Vision-language Pre-training for Enhanced CT Image Understanding. Shui et al. (2025).

---

> > > ### Comment · Reviewer_tLzt · 2025-08-06
> > >
> > > Thank you for your clarification and the added table.
> > >
> > > As I understand now, the primary reason that larger validation set results in worse performance in Table 3 is that the training set is smaller when the validation set is larger. If this is the case, then the necessary ablation would be GWA with a training set 90% (and also maybe 99%) of the side of the training set so that the training set size matches that of the val set. I would expect that GWA with only 90% training data would perform *worse* than the validation set result.
> > >
> > > Please let me know if this ablation already exists in the results (I may have missed it).

---

> > > > ### Author Response · Authors · 2025-08-06
> > > > **Further ablation as requested**
> > > >
> > > > Your intuition is correct. GWA with 90% training set and no validation set performs worse than using a 10% validation set for early stopping. The table below now includes this additional result.
> > > >
> > > > | | **CIFAR-10** | **CIFAR-N 9%** | **CIFAR-N 17%** | **ImageNet1k**
> > > > | :----------------------------- | ----: | ----: | ----: | ----: |
> > > > | **ViT - Val Set (90/10)** | 81.10 | 78.31 | 75.23 | 73.01 |
> > > > | **ViT - Val Set (90/1)**       | -0.07 | -0.01 | -0.20 | -0.08 |
> > > > | **ViT - GWA (90/0)**       | -0.12 | -0.01 | -0.27 | -0.21 |
> > > > | --- | --- | --- | --- | --- |
> > > > | **ConvNeXt - Val Set (90/10)** | 89.86 | 85.33 | 82.30 | 71.24 |
> > > > | **ConvNeXt - Val Set (90/1)**  | -0.01 | -0.02 | -1.12 | -0.18 |
> > > > | **ConvNeXt - GWA (90/0)**       |  -0.22 | -0.17 | -0.35 | -0.09 |
> > > >
> > > > Please note that the results of this ablation in the table above represent a sub-optimal scenario for both GWA and the 1% validation set, as the training set size has been artificially restricted.
> > > > In contrast, Table 3 in the manuscript shows the best-case scenario possible with each approach to fairly demonstrate the performance of all early-stopping criteria.
> > > > We will add this additional ablation to the appendix of the paper and clarify the discussion of this point.

---

> > > > > ### Comment · Reviewer_tLzt · 2025-08-06
> > > > >
> > > > > Thank you for providing this additional result. I will be increasing my rating accordingly.

---

### Official Review · Reviewer_YFoy · 2025-06-17

**Clarity:** 3
**Significance:** 2
**Originality:** 2
**Rating:** 4
**Confidence:** 4

**Summary:**

This paper introduces Gradient-Weight Alignment (GWA), a novel metric that measures the coherence between per-sample gradients and model weights. During training, the peak of this metric closely corresponds to the peak of test accuracy. Notably, GWA can serve as an effective early-stopping criterion even in the absence of a validation set. The authors demonstrate that early stopping using GWA outperforms both traditional validation set baseline and the recently proposed LabelWave method. Furthermore, the gradient alignment approach facilitates the assessment of training data quality: higher levels of noise in the data result in lower GWA scores and a broader distribution of alignment values. Individual alignment values also help to detect mislabeled training examples.

**Questions:**

1. Is GWA actually better than the early stopping criteria mentioned in the p. 1 Weaknesses?
2. Is it truly necessary to adjust the expected alignment by the excess kurtosis to obtain a reliable generalization proxy? The paper does not include any comparison plots between the behavior of GWA and the pure expected alignment $\mathbb{E}_i [\mathcal{A}_T]$.
3. Why do the authors measure the alignment of gradients with $w_T$ (the final weights) instead of $w_T - w_0$, where $w_0$ is the initial weight vector? Intuitively, aligning gradients with $w_T - w_0$ reflects whether per-sample gradients are coherent with the shift from the initialization point. A higher alignment would suggest meaningful learning is still occurring, while a lower alignment might indicate that optimization moves along a different direction, possibly signaling overfitting.
4. Could you please explain how the formula for the gradients of the final layer’s weights (given on line 153) is derived? Also, I understand that calculating gradients only for the final layer reduces the computational cost of GWA, but would the metric behave differently if gradients from all layers were considered?
5. Is it possible to apply GWA as an early stopping criterion in domains other than images and for tasks beyond classification?

**Ethical Concerns:**

["NO or VERY MINOR ethics concerns only"]

**Final Justification:**

I have decided to raise my score by +1 pt, as the authors addressed my concerns during rebuttal. More specifically, they have discussed several other methods I mention in my review, and Gradient Disparity was even included as a baseline. Moreover, several other questions were clarified (kurtosis correction, GWA for non-classification tasks, and allignment with $w_T - w_0$).

**Limitations:**

Although the authors clearly acknowledge the limitations related to the computation of GWA, they do not address what I consider a major shortcoming of the work: the limited number of baseline methods considered (see p. 1, Weaknesses).

**Paper Formatting Concerns:**

I do not have any formatting concerns.

**Quality:**

2

**Strengths And Weaknesses:**

**Strengths:**
1. The paper introduces an early-stopping metric that can be computed efficiently during training, without requiring a validation set.
2. The GWA metric also enables analysis of training data quality, such as detecting mislabeled examples.
3. Experiments cover both training from scratch and fine-tuning scenarios.
4. The paper is clearly written and easy to follow.

**Weaknesses:**
1. The experimental evaluation misses comparisons to several other methods that also explore stopping criteria without validation sets [1, 2, 3, 4, 5, 6]. In particular, [5], which measures disparity between different stochastic gradients, is closely related to GWA. These papers should be at least discussed in the related work section and ideally included as baselines.
2. The GWA metric appears overly complex, and the authors do not provide enough intuition or explanation for why this complexity is necessary (see questions for more details).
3. Although GWA seems broadly applicable across data types and training tasks, the experiments are limited to image classification only.

**References:**

[1] Maren Mahsereci, Lukas Balles, Christoph Lassner, and Philipp Hennig. Early Stopping without a Validation Set, 2017.

[2] Hwanjun Song, Minseok Kim, Dongmin Park, and Jae-Gil Lee. How does Early Stopping Help Generalization against Label Noise?, 2019.

[3] Hwanjun Song, Minseok Kim, Dongmin Park, Yooju Shin, and Jae-Gil Lee. Robust Learning by Self-Transition for Handling Noisy Labels, 2021.

[4] Yingbin Bai, Erkun Yang, Bo Han, Yanhua Yang, Jiatong Li, Yinian Mao, Gang Niu, and Tongliang Liu. Understanding and Improving Early Stopping for Learning with Noisy Labels, 2021.

[5] Mahsa Forouzesh and Patrick Thiran. Disparity Between Batches as a Signal for Early Stopping, 2021.

[6] Ali Vardasbi, Maarten de Rijke, and Mostafa Dehghani. Intersection of Parallels as an Early Stopping Criterion, 2022.

---

> ### Author Rebuttal · Authors · 2025-07-30
>
> > [...] evaluation misses comparisons to several other methods that also explore stopping criteria without validation sets [1, 2, 3, 4, 5, 6]. In particular, [5], which measures disparity between different stochastic gradients, is closely related to GWA. These papers should be at least discussed in the related work section and ideally included as baselines.
>
> Thank you for these references. We will address your points one-by-one.
> * [1] Gradient Signal: this work is closely connected to the idea of gradient alignment that we also leverage and that we discuss in our related work (see line 66 and following with [23] being most closely related). We will add this paper to the discussion and want to point out that we actually compare GWA against similar work in Appendix A.2. While we found pairwise gradient alignment and gradient SNR to correlate with GWA, the approaches were noisier and more importantly, their associated compute overhead made it prohibitively expensive to run on the larger datasets and models we evaluate in this manuscript.
> * [2, 3, 4]: Learning with Label Noise: While not the primary focus of our work, we consider this line of work important and choose LabelWave as the primary representative baseline due to its recency and strong performance in the field. While effective in their respective regime, these approaches are often not sensitive enough to subtle overfitting, e.g. in robust models such as ConvNeXt (see Figure 2 center). This is a key limitation which GWA addresses.
> * [5] Batch Gradient Comparison: As per your suggestion, we have now run an additional experiment comparing GWA to the gradient disparity method from [5], which computes the L2 distance between successive batch gradients instead of per-sample gradients. We find that gradient disparity fails completely on our benchmarks (note that the most sophisticated experiment in [5] is a CIFAR-100 subset with 1.28k samples), while its computation also incurs substantial overhead (see also Table 4 in [5]) by requiring to store the gradients across multiple steps to compute the metric. Moreover, and unlike GWA, it does not provide any per-sample insights.
> * [6] Parallel Instances of Linear Models: This work proposes to train and compare two parallel instances of a linear model to determine early stopping. While this primarily theoretical work additionally proposes an extension to multi-layer networks, we could not find a suitable way to implement this for any current architecture or dataset evaluated in our work because the work provides no results on e.g. convolution or attention layers.
>
> | CIFAR-10-N             | 0%     | 9%     | 17%    |
> | ---------------------- | ------ | ------ | ------ |
> | Val Set 10%            | 89.86% | 85.33% | 82.30% |
> | GWA                    | 89.73% | 86.08% | 82.55% |
> | Gradient Disparity [5] | 55.85% | 57.93% | 53.85% |
>
> *Table R1: Test accuracy achieved with a ConvNeXt on CIFAR-10-N with different levels of label noise when selecting the optimal model with early stopping based on Gradient Disparity [5], the standard 10% validation set, and our proposed GWA method.*
>
>
> We will incorporate a discussion of these approaches in the camera ready version.
>
> > The GWA metric appears overly complex, and the authors do not provide enough intuition or explanation for why this complexity is necessary. [...] Is it truly necessary to adjust the expected alignment by the excess kurtosis to obtain a reliable generalization proxy? The paper does not include any comparison plots between the behavior of GWA and the pure expected alignment $\mathbb{E}_i[\mathcal{A}_t]$.
>
> We agree that a more intuitive explanation is beneficial and will add it to the paper. The core idea of GWA is straightforward: measuring the alignment between per-sample gradients and model weights. The perceived complexity in Eq. (2) and (3) stems from the explicit statistical formulation for computing the moments of the alignment distribution, which are essential for a robust and computationally efficient online estimator. Seeing as a rigorous definition is necessary, this is an instance where we chose clarity over simplicity.
>
> Yes, the kurtosis adjustment is crucial. Relying solely on the mean alignment, $\mathbb{E}[\mathcal{A}_t]$, is insufficient because the mean does not capture the shape of the alignment distribution, specifically its tailedness. This property, measured by kurtosis, reflects the disproportionate influence of atypical samples, a key factor in generalization according to the long-tail theory (see [12, 13] in our submission). While the mean indicates the central tendency of alignment, our kurtosis-based correction allows the metric to better “see” these influential outliers. On datasets with noisy labels (e.g., CIFAR-10-N), where a subset of atypical samples is memorized late in training, the mean-only estimator is sensitive to these tails and provides a misleading early-stopping signal that continues to change after generalization performance has peaked.
> We will add an ablation plot contrasting GWA with the uncorrected mean $\mathbb{E}[\mathcal{A}_t]$ to the final version to clearly demonstrate this necessity for kurtosis correction.
>
> > Although GWA seems broadly applicable across data types and training tasks, the experiments are limited to image classification only. Is it possible to apply GWA as an early stopping criterion in domains other than images and for tasks beyond classification?
>
> Yes, GWA is broadly applicable, and its computational efficiency is designed for scalability across domains. GWA extends directly to other settings:
> * Self-Supervised Learning: GWA is applicable to contrastive learning methods where the InfoNCE loss functions can be seen as a multi-class cross-entropy task. We deferred a full analysis to maintain the paper's focus on clean per-sample attribution, which is more complex when gradients depend on the set of negative samples.
> * NLP: GWA's applicability to autoregressive language modeling is more straightforward, as their training objective is a cross-entropy loss over the vocabulary. The challenge is not in applying GWA but in interpreting the resulting token-level dynamics, which we identify as a promising direction for future work.
>
> Please also refer to our response to reviewer RSWb on a related point.
>
> > Why do the authors measure the alignment of gradients with $w_t$ (the final weights) instead of $w_T - w_0$, where $w_0$ is the initial weight vector?
>
> The reason for not using the vector $w_t - w_0$ is that this vector measures the displacement from initialization. Thus, aligning gradients with this vector would assess if learning continues along the path from initialization rather than towards convergence. This is not a known indicator of generalization. Conversely, our choice to align with $w_t$ is directly motivated by theoretical work [2, 25] on directional convergence, which shows that gradients of well-generalized samples align with the *current* weight vector $w_t$. This provides a theoretically-grounded measure, which we also show empirically (Appendix, Fig. 9) to be an effective online proxy for the alignment with “optimal” weights $w^*$ (proposed by Guille-Escuret et al. (2024) - see [28] in our submission).
>
> > [...] explain how the formula for the gradients of the final layer’s weights (given on line 153) is derived? [...] would the metric behave differently if gradients from all layers were considered?
>
> The formula on line 153 is the standard closed-form gradient for a linear layer with a softmax cross-entropy loss. It is derived using the chain rule and represents the outer product of the latent representation $z_i$ and the prediction error $\hat{y}_i - y_i$, a technique also leveraged by prior work [29].
>
> Regarding the use of full versus final-layer gradients: yes, the metric would behave differently. It would become computationally more complex and noisier. The final layer’s gradient provides the most direct and stable signal of the learned task, as a classifier’s goal is to learn a representation that is linearly separable by this layer [R3]. Conversely, gradients from earlier layers are more unstable [R4] and an indirect signal [R3]. This instability, coupled with the high dimensionality of full gradients, leads to a metric that trends towards zero, obscuring the generalization signal.
> We have conducted an ablation study showing that: (1) including earlier layers produces a noisier metric, leading to premature and suboptimal early stopping; and (2) the alignment signal from earlier layers is significantly weaker (closer to zero) than the final layer's signal. We will include this in the camera-ready version.
>
> [R3] Understanding intermediate layers using linear classifier probes. Alain and Bengio (2016).
>
> [R4] Evaluating the Stability of Semantic Concept Representations in CNNs for Robust Explainability. Mikriukov et al. (2023).

---

> > ### Comment · Reviewer_YFoy · 2025-08-01
> > **Raising score to 4 (borderline accept)**
> >
> > I acknowledge that I have read the response from the authors. I also would like to thank the authors for the thorough rebuttal. Given that my concern with an insufficient number of baselines was resolved and my questions were clearly answered, I have decided to upgrade my score by +1 pt. I hope that the clarifications provided by the authors during rebuttal will be included in the final version of the manuscript, in case the paper is accepted.

---

> > > ### Author Response · Authors · 2025-08-04
> > > **Thank you to the reviewer**
> > >
> > > Thank you for your constructive feedback! We will include the suggested clarifications in the manuscript.

---

### Official Review · Reviewer_CWdC · 2025-06-21

**Clarity:** 3
**Significance:** 3
**Originality:** 2
**Rating:** 4
**Confidence:** 4

**Summary:**

This paper proposes a metric - *Gradient-Weight Alignment* (GWA) that aims to provide an understanding of the training dynamics and its impact on generalization. Specifically, the paper explores utilizing GWA for early stopping. The paper also presents the robustness of GWA by utilizing the early stopped model on a perturbed test set. Experiments are performed on standard computer vision datasets and ViT and ConvNeXt models to asses the metric.

**Questions:**

To improve the paper's clarity and rigor, please address the following:
1. How is the per-sample alignment score $\gamma(x_i, w_T)$ handled numerically when the gradient $g_T(x_i)$ is a zero vector, which should occur for samples that are perfectly classified?
2. Could you provide a quantitative comparison of the computational complexity and runtime overhead of GWA versus LabelWave and standard validation?
3. How robust is GWA to different model initializations and variations in model size within the same architecture (e.g., ViT-Small vs. ViT-Base)? The paper claims to investigate this but provides no direct evidence.
4. What data augmentation transforms were used during training? Could these transforms, especially aggressive ones, affect the stability and interpretation of the GWA metric?
5. The rationale for Figure 3 is not entirely clear. While it's expected that lower data noise leads to better accuracy and alignment, how does this demonstrate the reproducibility or consistency of the metric itself? Please clarify the intended message.
6. Could you provide more intuition on why the higher-order moments in GWA (i.e., the kurtosis correction) make it more sensitive to overfitting than traditional validation or methods like LabelWave, as suggested by the results in Figure 2?

**Ethical Concerns:**

["NO or VERY MINOR ethics concerns only"]

**Final Justification:**

The authors have clarified many of my concerns as well as concerns raised by the other reviewers.

**Limitations:**

The authors state in the checklist that limitations are discussed, but this is not the case. The paper lacks a dedicated discussion of its limitations. Key limitations to address include the potential numerical instability of the GWA metric (the zero-gradient issue), the strong assumption that last-layer gradients are a sufficient proxy for full-model gradients, and potential failure modes of the method.

**Paper Formatting Concerns:**

- The paper claims results are averaged across triplicate runs, but the tables in the main text lack error bars or standard deviations, which are essential for assessing statistical significance. These should be included.
- The details of the perturbations for CIFAR-C/ImageNet-C in Table 2 could be specified more clearly in the main text for better readability, rather than only citing the benchmark paper.

**Quality:**

2

**Strengths And Weaknesses:**

**Strenghts**:
- The primary strength of this work is that it proposes a validation-set-free method for monitoring training and selecting a model for deployment. This is a significant practical advantage, particularly in data-scarce scenarios where withholding a validation set is costly. The method provides a promising alternative to the standard practice of using a hold-out set.
- The paper addresses the important and challenging problem of understanding the training dynamics of deep neural networks without relying on computationally expensive second-order methods, such as Hessian-based analysis.

**Weaknesses**:

However, it faces a few issues that need to be addressed before being ready for publication.
- **Insufficient Comparison and Positioning with Relevant Work:**
    - The paper should be more clearly positioned with respect to Training Data Attribution (TDA) methods and influence functions. While some works are cited , the paper dismisses them as post-training analysis  without a deeper comparison. More recent and relevant methods like Datamodels [1] and approximate unrolling [2] are not discussed.
    - The methodology, which relies on the alignment of gradients with the final layer weights, is closely related to the well-established phenomenon of Neural Collapse [3]. This connection is a critical omission. The paper should explicitly discuss this relationship and differentiate GWA from other metrics derived from Neural Collapse properties, such as the information-theoretic HDR and MIR metrics [4], which also measure alignment to analyze generalization.

- **Unresolved Issues with the Metric's Formulation:**
    - The definition of the per-sample alignment score, $\gamma(x_i, w_T)$, becomes problematic when a sample is perfectly classified. For a cross-entropy loss, a perfect classification results in a gradient that is zero or near-zero, making the cosine similarity term $\frac{g_T(x_i) \cdot w_T}{\Vert g_T(x_i)\Vert \Vert w_T \Vert}$ undefined due to a zero-norm gradient vector. This is a fundamental issue, as many samples will be learned perfectly during training. The paper's claim that alignment approaches 1 (line 106) for perfectly classifiable data  seems to overlook this practical and theoretical problem.
    - This concern is reinforced by the paper's own efficient gradient calculation, in line 153, which explicitly shows the gradient becomes a zero vector if $\hat{y}_i=y_i$.

- **Claims Not Substantiated by Experimental Evidence:**
    - The paper repeatedly claims that GWA is computationally efficient and scalable. However, no empirical evidence (e.g., wall-clock time, FLOPS) is provided to compare its computational cost against the baselines it aims to replace or improve upon, such as validation set evaluation or LabelWave
    - In Section 4.2, the paper states it will "investigate whether GWA remains consistent across multiple model initializations". However, the experiments only report results averaged over three runs, with no analysis or visualization of the variance or consistency across these different initializations.

**References**

[1] *Ilyas, Andrew, et al. "Datamodels: Predicting Predictions from Training Data." Proceedings of the 39th International Conference on Machine Learning. 2022.*

[2] *Bae, Juhan, et al. "Training data attribution via approximate unrolling." Advances in Neural Information Processing Systems 37 (2024): 66647-66686.*

[3] *Papyan, Vardan, X. Y. Han, and David L. Donoho. "Prevalence of neural collapse during the terminal phase of deep learning training." Proceedings of the National Academy of Sciences 117.40 (2020): 24652-24663.*

[4] *Song, Kun, et al. "Unveiling the Dynamics of Information Interplay in Supervised Learning." International Conference on Machine Learning. PMLR, 2024.*

---

> ### Author Rebuttal · Authors · 2025-07-30
>
> > Insufficient Comparison and Positioning with Relevant Work
>
> Thank you for your feedback. As you recommended, we will incorporate this discussion into the paper. Concretely:
>
> *Regarding TDA*: Our related work already discusses foundational Training Data Attribution (TDA) and influence function estimation methods, primarily to distinguish GWA as an efficient, online metric. While we agree that the works you recommended ([1], [2] of your references) are valuable in their own right, they do not resolve the core issues stated in our related work, mainly the computational overhead:
> * Datamodels: The framework's core requirement is to train a large number of models from scratch, e.g., **300k models** for a ResNet9 for CIFAR in Table 1 of [1].
> * Approximate Unrolling with SOURCE: The “efficient” version of SOURCE “is L [constant factor of 2 or 3] times **more computationally expensive than influence functions**”, while standard SOURCE requires the computation of the EK-FAC at **every** checkpoint (see Section 3.4 in [2]).
>
> Nonetheless we recognise the importance of these methods. Our original submission already included TracIn [R5] (based on gradient norm), an influence function approximation (Appendix A.2). We have now additionally evaluated second-order derivative (curvature-based) influence directly via a block-diagonal Hessian approximation, as is common. We found these first- and second-order metrics to be highly correlated. However, critically, both fail to provide a reliable early-stopping signal, whereas GWA succeeds at this key objective of our work. For instance, TracIn's signal is inconsistent for ConvNeXt and ViT models despite the models having comparable validation accuracy (Figure 8 right). We will add this new analysis to the camera-ready manuscript.
>
> *Regarding Neural Collapse (NC)*: While NC [3] is an important theoretical phenomenon which describes the terminal-phase collapse of last-layer features to their structured class-means, GWA is fundamentally different. GWA analyzes the per-sample gradient-weight interaction throughout training on realistic, noisy data, where quantifying variance is the key signal for generalization.
>
> However, regarding the specialized case of MIR/HDR: This is an insightful connection and we thank you for pointing this out. MIR/HDR [4] use features for a global, dataset (or batch)-level perspective on alignment. In contrast, GWA uses per-sample gradients, offering a more granular, complementary view. We have now also evaluated MIR and HDR, but found that the computational requirements of the method (which are also mentioned in Section 5 in [4]) are so high that neither method is a suitable alternative for GWA. In fact, even the small scale batch-approximations proposed in [4] are too computationally expensive to compete with GWA (we had to reduce the batch size on CIFAR-10 on a 48GB GPU). Nonetheless, the values of MIR seem to correlate with GWA in our preliminary investigation. We will add these results to the camera-ready version and believe further connecting these two approaches in future work is a promising way to integrate GWA and information-theoretic research.
>
> > The definition of the per-sample alignment score, $\gamma$, becomes problematic when a sample is perfectly classified. For a cross-entropy loss, a perfect classification results in a gradient that is zero or near-zero, making the cosine similarity term $\text{cos sim}$ undefined due to a zero-norm gradient vector. [...] concern is reinforced by the paper's own efficient gradient calculation, in line 153, which explicitly shows the gradient becomes a zero vector if $\hat{y_i} = y_i$.
>
> Thank you for this insightful question. You are correct, that in theory, the gradient $g_T$ becomes zero if $\hat{y}_i = y_i$. However, the logits will never match the true one-hot class label $y_i$, even for perfectly classifiable data, as the softmax output probabilities only asymptotically approach the true class labels. In practice, the per-sample alignment will never reach 1, as we do not have perfectly classifiable data. Thus, the asymptotic theoretical insight in line 106 only serves as intuition for the range of alignment scores. If you believe this to be confusing for the reader, we are happy to remove it. Please note additionally that we also add a small epsilon to the denominator for numerical stability, as is standard practice.
>
> > The paper repeatedly claims that GWA is computationally efficient and scalable. However, no empirical evidence (e.g., wall-clock time, FLOPS) is provided [...] Could you provide a quantitative comparison of the computational complexity and runtime overhead of GWA.
>
> When training a ViT/S-16 implemented in JAX on ImageNet1k with a single NVIDIA RTX A6000, GWA adds $\sim 2.5$sec to the per-epoch wall-clock time (on average 1861 images/s with GWA vs. 1867 images/s without GWA for $224^2$px). This is more efficient than evaluating a 1% validation set ($\sim 16$sec overhead for one iteration). Computing the closed-form gradients and the cosine similarity requires $\sim 0.003$ GFLOPs compared to 4.6 GFLOPs of a single forward pass with a ViT/S-16. Peak GPU memory when using active deallocation in this setting is 25.11GB with or without GWA (no difference). Thus, GWA has minimal overhead. We will add these quantitative results to the paper as you requested.
>
> > [...] experiments only report results averaged over three runs, with no analysis or visualization of the variance or consistency across these different initializations.
>
> > How robust is GWA to different model initializations and variations in model size within the same architecture (e.g., ViT-Small vs. ViT-Base)?
>
> > The rationale for Figure 3 is not entirely clear.
>
> On Consistency: We already provide run-to-run variance (min-max deviation) for all methods in Appendix, Tables 5 and 6, showing that GWA's stability is comparable to or better than baselines.
>
> On Robustness: While absolute GWA values can differ across model architectures and sizes (e.g., ViT-S vs. ConvNeXt), its strong positive correlation with final performance remains consistent, which is the key property, as shown in Figure 3 of the manuscript. We will additionally include quantitative results for different model sizes, which are consistent with the results in Figure 3 (i.e. lower performance correlates with lower alignment independent of model size).
>
> On Figure 3 Rationale: The figure's purpose is to demonstrate GWA's reliability for model comparison. Each point is a distinct run (varied initializations, noise levels, architectures). The strong correlation proves that GWA consistently ranks models: a **higher peak GWA value reliably predicts higher final test accuracy**, validating its use for model selection across different training configurations.
>
> > What data augmentation transforms were used during training? Could these transforms, especially aggressive ones, affect the stability and interpretation of the GWA metric?
>
> All experimental details, including augmentations (RandAug, Mixup), can be found in Table 4 in the Appendix. Empirically, we find that there is no negative impact on GWA when using augmentations. Rather, the model trains more consistently, learning better generalizing features, and resulting in higher GWA.
>
> > [...] more intuition on why the higher-order moments in GWA (i.e., the kurtosis correction) make it more sensitive to overfitting than traditional validation or methods like LabelWave, as suggested by the results in Figure 2?
>
> Regarding the use of higher order moments: A mean-only estimator is only sufficient for concentrated, unimodal distributions. This assumption is inconsistent with both the long-tailed theory of sample influence ([12, 13] in the paper) and our empirical observations (exemplarily Figure 2 right). Kurtosis provides a robust correction for this deviation that aligns with the long-tail theory. Note that we do not claim that the Kurtosis correction is the main reason for the strong capability of GWA in detecting overfitting, but rather the per-sample granularity that captures potential long-tail dynamics. LabelWave and other methods summarize the contributions of individual samples, e.g. by averaging, while we leverage the entire distributional view.
>
> > Paper Formatting Concerns
>
> As you recommended, we will elaborate more on the perturbations of the CIFAR-C and ImageNet-C dataset within the paper and will address the other formatting concerns in the updated manuscript.
>
> [R5] Estimating training data influence by tracing gradient descent. Pruthi et al. (2020).

---

> ### Comment · Reviewer_CWdC · 2025-08-01
> **Response to Authors**
>
> I thank the authors for clarifying many of my concerns, as well as the concerns raised by other reviewers.
>
> I would recommend highlighting the points regarding the formula, as well as adding the computational efficiency discussion posted here in the paper (at least in the appendix if there is no space in the main paper).
>
> Since most of my questions have been answered, I will raise my score to a borderline accept.
>
> Additionally, do the authors have any insight into why TracIn is more unstable than the metric proposed?

---

> > ### Author Response · Authors · 2025-08-04
> > **Thank you to the reviewer**
> >
> > Thank you for the productive discussion! Regarding your point about TracIn, this is an interesting question. We hypothesize TracIn's inconsistency is due to its strong dependency on the gradient magnitudes. We observe that models like ConvNeXt exhibit particularly high and volatile gradient norms in early training compared to ViT.
> > GWA does not suffer from this issue while using the model gradients due to the inherent normalization within the cosine similarity formula. This isolates the influence of gradient direction from that of magnitude, making our metric robust to these fluctuations in gradient norms.
> >
> > We will include the points from our discussion on GWA’s formulation and the computational efficiency in the manuscript as you suggested.

---

> > > ### Comment · Reviewer_CWdC · 2025-08-04
> > > **Final response**
> > >
> > > Thank you for the thoughtful observation. It's a very interesting point. While this may fall slightly outside the current scope of the paper and isn't necessarily a limitation, I encourage the authors to consider including it, time permitting. Even a brief inclusion with some ablations would add valuable context.
> > >
> > > Given the widespread use of TracIn and its variants across many applications, demonstrating how and why this proposed metric performs better (even in specific scenarios) could greatly enhance the impact and visibility of your work.
> > >
> > > If you already have any preliminary results along these lines, I’d love to see them; it would certainly strengthen my support for the paper. Of course, I understand if time constraints make this difficult.

---

> > > > ### Author Response · Authors · 2025-08-06
> > > > **Thank you**
> > > >
> > > > Thank you for this suggestion. We agree that including a more in-depth comparison to TracIn will further strengthen our paper.
> > > > Regarding our previous discussion, we assume you would like us to include ablations on the point of gradient norm volatility. We have this data available and will include it in the appendix and reference it in the main manuscript. Exemplarily, to illustrate the gradient norm dependency of TracIn: in Figure 8 right, bottom panel, the distribution of the gradient norms that are used to compute TracIn is extremely skewed and dominated by outliers (with magnitudes up 7160 at its peak at epoch 12). In comparison, GWA remains unaffected. Its distribution is, by definition, compactly supported and remains concentrated with the gradient norm outliers not influencing it since GWA depends on direction only. We are unfortunately unable to include a figure in this rebuttal but will add a detailed analysis with figures to the camera-ready version.
> > > >
> > > > On your follow-up question: A specific scenario where we foresee GWA to be particularly useful is in large-scale training data attribution, where, due to its high efficiency, it can be used either as a replacement or in conjunction with other training data attribution methods like TracIn. Note that, while TracIn is popular due to its low computational overhead, GWA is even more efficient as it does not require full model per-sample gradients over several checkpoints. Using GWA as a training data attribution technique is something we are currently actively exploring. Our correlation results in Figure 8 indicate that both metrics allow identifying more or less influential samples. Moreover, the interplay of both TracIn and GWA can be leveraged to identify distinct clusters that, in our preliminary results, seem to correspond to easy- vs. hard-to-fit samples (compare also the brief discussion in lines 502 and following).
> > > > Thank you again for the thorough review and your constructive suggestions!

---

> > > > > ### Comment · Reviewer_CWdC · 2025-08-07
> > > > > **Regarding TracIn Comparison**
> > > > >
> > > > > Thank you for providing some more insight on this. It would definitely be a good addition to the paper!

---

### Official Review · Reviewer_iv9T · 2025-06-27

**Clarity:** 3
**Significance:** 4
**Originality:** 2
**Rating:** 5
**Confidence:** 3

**Summary:**

Previous work has argued that the alignment of gradients and model parameters is an indicator of convergence of a deep neural network (DNNs). This paper proposed an efficient estimator for this alignment by measuring the alignment per sample, and shows empirically that it can serve as a useful signal when determining when to halt optimization (early stopping) and selecting influential training samples. The estimator is made efficient by focussing on the gradients of the last layer, and computing it in batches (during an epoch).

**Questions:**

The inclusion of kurtosis in the estimator is motivated by the idea that rare/atypical samples have an outsized influence on the model, and that the kurtosis can account for this. Do the authors also have empirical support for this argument? It would be good to show that if one uses simply the first moment, that the estimator becomes worse for certain objectives (for instance in terms of determining when to halt optimization).

**Ethical Concerns:**

["NO or VERY MINOR ethics concerns only"]

**Final Justification:**

The authors convincingly addressed the weaknesses listed as well as my remaining question. While their response improves my assessment of the paper I do not believe this warrants an increase in the rating.

**Limitations:**

Yes

**Paper Formatting Concerns:**

None found.

**Quality:**

3

**Strengths And Weaknesses:**

I believe this paper studies an important research question (how can we measure the generalization of a DNN without a validation set) and makes an clear contribution to it. It contains an extensive set of experiments to empirically support the usefulness of the introduced estimator. I believe the paper is fit for acceptance, although it still has some weaknesses in terms of the theoretical support for the estimator.  Specifically the design choices to focus on the last layer and computing it in batches.
1. The authors write: *'we exploit the fact that classification fundamentally operates on latent representations. This reduces irrelevant variations'*. While somewhat intuitive, it is not entirely clear to me why the gradient alignment in layers previous to the last layer constitutes of irrelevant variations. It would be good if the authors could either show some connection between alignment in the last layer and alignment in all other layers (which would justify focussing on it), or show in a small-scale experiment that little is gained from measuring alignment on all layers.
2. The authors write compute the alignment per batch, while in between batches the parameters. I understand that this has an advantage in terms of computational resources, but it would be good to know what the costs are in terms of accurately estimating the alignment. Particularly, it would be good to theoretically analyse how the difference between $E[A_T]$ and $E[\hat{M}_T^1]$ (the bias of the estimator for the first moment) depends on the learning rate. Intuitively, if there are big steps in between batches, this bias will increase. **To be clear**: I do not think this makes the estimator not useful in practice, but it would be good to have an idea about the conditions under which it yields a good estimate of the alignment.

---

> ### Author Rebuttal · Authors · 2025-07-30
>
> Thank you for your positive assessment and your insightful questions.
>
> > [...] it is not entirely clear to me why the gradient alignment in layers previous to the last layer constitutes of irrelevant variations. It would be good if the authors could either show some connection between alignment in the last layer and alignment in all other layers (which would justify focussing on it), or show in a small-scale experiment that little is gained from measuring alignment on all layers.
>
> Our focus on the final layer is a principled design choice, grounded in both theory and practical considerations.
>
> A deep classifier's primary goal is to learn a representation that is linearly separable by its final layer [R3]. Consequently, last-layer gradients provide the most direct, stable signal of the learned task. Gradients in preceding layers, while necessary for building the representation, are an indirect signal; their variations are "irrelevant" as the corresponding information is condensed in the final representation [R3].
>
> In fact, for our case, including earlier layers degrades the estimator. Gradients in shallower layers are significantly more unstable, as shown in work such as [R4]. Together with the high dimensionality of full model gradients, this can lead GWA to go in expectation towards zero, rendering the metric statistically uninformative. On the other hand, the final layer's gradients are lower-dimensional and more directly structured by the classification task, providing a robust signal for evaluating model performance.
>
> On your recommendation, we have conducted an ablation study showing that: (1) including earlier layers produces a noisier metric, leading to premature and suboptimal early stopping; and (2) the alignment signal from earlier layers is significantly weaker (closer to zero) than the final layer's signal. We will include this in the camera-ready version.
>
> > [...] it would be good to know what the costs are in terms of accurately estimating the alignment.  Particularly, it would be good to theoretically analyse how the difference between $E[\mathcal{A}_t]$ and $E[\hat{\mathcal{A}}_t]$ (the bias of the estimator for the first moment) depends on the learning rate. Intuitively, if there are big steps in between batches, this bias will increase.
>
> Your intuition is absolutely correct.
> The bias of our estimator is indeed proportional to the learning rate $\eta$. A first-order Taylor expansion of the alignment function $\mathcal{A}(w_t)$ around the epoch's initial weights $w_0$ shows the expected bias is:
> $\text{Bias} \approx \frac{1}{T}\sum_{t=0}^{T-1} \mathbb{E}\left[\nabla_w \mathcal{A}(w_0)^{\top}(w_t - w_0)\right]$
> Since the weight drift $(w_t - w_0)$ is a sum of $t$ gradient updates scaled by $\eta$, the bias is approximately linear in $\eta$. For our cosine similarity metric, the bias is driven by the change in the weight vector's direction.
> We will include this in the manuscript but also already have validated this empirically. In Figure 6 (Appendix), our online estimator and an offline ground-truth calculation yield closely matching curves, validating your intuition that the estimator remains useful.
>
> > [...] inclusion of kurtosis in the estimator is motivated by the idea that rare/atypical samples have an outsized influence on the model, and that the kurtosis can account for this. Do the authors also have empirical support for this argument? It would be good to show that if one uses simply the first moment, that the estimator becomes worse for certain objectives (for instance in terms of determining when to halt optimization).
>
> Yes, we can confirm this empirically. On clean datasets like CIFAR-10, both estimators perform comparably. However, on datasets with noisy labels (e.g., CIFAR-10-N), where a subset of atypical samples is memorized late in training, their behavior diverges. In this setting, the atypical samples create a heavy-tailed alignment distribution. The mean-only estimator provides a misleading signal in this scenario, with the mean continuing to change after generalization performance has peaked leading to an incorrect early stopping signal. In contrast, our kurtosis-corrected estimator is robust to these outliers and better tracks the validation accuracy, making it a more reliable indicator for halting optimization.
>
> We have conducted an ablation directly comparing mean-only and kurtosis-corrected GWA against validation performance over time on CIFAR-10-N. We will add these results, as well as visualizations of the alignment distribution and correlation of both metrics, to illustrate the key advantage of kurtosis correction.
>
>
> [R3] Understanding intermediate layers using linear classifier probes. Alain and Bengio (2016).
>
> [R4] Evaluating the Stability of Semantic Concept Representations in CNNs for Robust Explainability. Mikriukov et al. (2023).

---

> > ### Comment · Reviewer_iv9T · 2025-07-31
> >
> > I acknowledge that I have read the response from the authors. I want to thank them for conducting additional experiments based on my initial review, and appreciate their effort. Below, I will briefly go over each discussed issue below.
> >
> > 1. While I somewhat disagree that the gradient of the earlier layers is ‘irrelevant’ (despite the argument of [r3]), I now better understand the design choice of the authors to focus on the final layer,  and believe this weakness to be resolved. I think the paper would benefit from including part of this explanation.
> >
> > 2.  It would be useful to add the derivation of this approximate bias to the manuscript (and if the word limit allows it, in a reply below). It would also be insightful to recreate Figure 6 from the appendix for different learning rates to show that the bias is increasing  linear in the learning rate (e.g. as the learning rate increases, the offline and online estimator should diverge if what you say is correct).
> >
> > 3. I appreciate the explanation provided by the authors. In addition to the experiments mentioned, it would be good to include this explanationas a motivation for the inclusion of the kurtosis metric.
> >
> > While the additional experiments improve my assessment of the paper, they do not prompt an improvement in the rating. I believe the submission is fit for acceptance.
> >
> > [r3] Understanding intermediate layers using linear classifier probes. Alain and Bengio

---

> > > ### Author Response · Authors · 2025-08-04
> > > **Thank you to reviewer**
> > >
> > > Thank you for your constructive feedback. The following changes will be made to the camera-ready version:
> > >
> > > * The discussion on early-layer gradients will be incorporated into the manuscript, and we will revise the term ‘irrelevant’ as we agree that this can potentially lead to confusion.
> > > * The complete derivation for the estimator bias will be added to the appendix. As suggested, we have now also empirically evaluated how the bias changes across a larger range of learning rates. Indeed, as you assumed, the learning rate is strongly linearly correlated with the absolute bias (Pearson R $\approx 0.9$). We will add a figure illustrating this relationship as requested.
> > > * Moreover, we will integrate the motivation for including the kurtosis as a correction factor into the main paper following your feedback.

---

### Official Review · Reviewer_RSWb · 2025-07-03

**Clarity:** 3
**Significance:** 2
**Originality:** 3
**Rating:** 3
**Confidence:** 4

**Summary:**

This paper introduce a new metric called gradient-weight alignment that can be considered as a novel proxy for generalization performance during training, which helps to indicate the most generalized model during training, especially with noisy data. This paper first show per-sample gradient scores aligns well with the generation performance during the training trajectory. The motivation is that the gradient would consistently point in the same direction as the model weights given enough training. However, calculate the per-sample grandient alignment is not cheep given that the neural networks are usually quite heavy and with grate amount of training data. As a result, this paper introduce an estimate that using the alignment with the last linear layer and accumulate the scores during the training epochs. This paper shows that for image classification task, GWA can be a good indicator and achieve better performance on noisy dataset.

**Questions:**

See above

**Ethical Concerns:**

["NO or VERY MINOR ethics concerns only"]

**Limitations:**

See above

**Paper Formatting Concerns:**

No.

**Quality:**

2

**Strengths And Weaknesses:**

The paper is easy to follow and well motivated. The experiment results are good especially for the noisy training dataset and finetuning tasks.  It seems that this method has very strong limitation since it has to be used for classification models due to the size of the weight. However, is it possible to extend this method to other embedding models? For example, self-supervised tasks such as contrastive learning. Also, does this method hold for other optimizer or training strategy such as sharpness-aware-minimization or adversarial training. For classification tasks, we also have many other optimization settings, such as few-shot learning, meta-learning. Or even many NLP tasks are based on the embeddings to get the prediction in the token space. For those cases, will it still hold? The cases this paper shows are quite limited.

---

> ### Author Rebuttal · Authors · 2025-07-30
>
> Thank you for your positive feedback and insightful questions about the broader applicability of our method.
> You're right that our current evaluation focuses on classification, but GWA is not limited by model size or task type. In fact, it’s fundamentally optimizer-agnostic and its adaptation to most common tasks is straightforward. We chose the classification setting to rigorously establish our core claims, given its direct link to the motivating theory ([2, 25] in the manuscript).
>
> In particular, on extending GWA to other tasks and settings:
> * For Self-Supervised Learning: Yes, GWA is directly applicable. The InfoNCE loss is structurally a multi-class cross-entropy loss. The model classifies the correct positive pair against a set of negatives. A closed form computation of the per-sample gradients for the estimator is possible. However, sample-level attribution is more complex due to the inherent dependency on negatives. We thus deferred a detailed analysis to maintain this focus.
> * Other Optimizers and Training Strategies: GWA works without modification. It uses raw gradients, so it’s independent of the optimizer’s update rule. Our results already show robustness across Adam, AdamW, SGD (Appendix, Tab. 4) for common training settings. As GWA is widely applicable, there is a range of different tasks it can be applied to, with few-shot learning being an interesting future direction.
> * For NLP: GWA is directly applicable to autoregressive models, as token prediction is cross-entropy over a vocabulary, so the same framework applies. In preliminary experiments, we found that GWA remains computationally tractable even when the final classifier layer is large (i.e., large vocabulary). However, it’s important to note that interpreting token-level dynamics in NLP over a single pass of the dataset results in intrinsically different training dynamics compared to common computer vision tasks. This warrants a more sophisticated investigation that we have planned in future work (see e.g. memorization in computer vision vs. language [R1, R2]).
>
> [R1] What Neural Networks Memorize and Why: Discovering the Long Tail via Influence Estimation. Feldman and Zhang (2020).
>
> [R2] Counterfactual Memorization in Neural Language Models. Zhang et al. (2023).

---

> ### Author Response · Authors · 2025-08-06
> **Gentle ping**
>
> Dear Reviewer RSWb,
>
> We would appreciate if you could engage with our rebuttal and let us know if your concerns have been addressed so that we can use the remaining reviewer-author discussion period to adress any outstanding points.
>
> Thank you in advance!

---

> ### Comment · Area_Chair_AuwZ · 2025-08-07
> **Friendly Reminder: Please Check Rebuttal Before Discussion Deadline**
>
> Dear Reviewer RSWb,
>
> Just a kind reminder to take a moment to review the authors’ rebuttal and see whether it addresses your concerns, as the discussion deadline is approaching.
>
> Thank you again for your thoughtful contributions to the review process.
>
> Best regards,
>
> AC AuwZ

---

### Note · Authors · 2025-08-12

We thank the reviewers and AC for a very constructive discussion phase that led to new experiments and analyses which we believe have strengthened our manuscript. We are pleased that all engaged reviewers recognised these efforts, leading to a positive resolution of their concerns and increases in their scores.

- With **Reviewer iv9T**, we clarified (1) the grounding and empirical evidence (new ablation) for the last-layer focus, (2) validated the estimator bias (approx. linear in LR), and (3) discussed the role of kurtosis correction for robustness in noisy settings (new ablation).
- With **Reviewer CWdC**, we discussed our positioning regarding TDA and Neural Collapse (including a new MIR/HDR analysis showing GWA is complementary and far more efficient), clarified numerical stability (zero-gradient issue), and provided quantitative evidence of GWA’s efficiency (<1% overhead). We also had a productive follow-up discussion on why GWA is more stable than TracIn (normalized direction vs. magnitude volatility).
- In response to **Reviewer YFoy**, we included/discussed new baselines (e.g., Gradient Disparity, which GWA substantially outperforms—Table R1), justified the kurtosis correction, and clarified GWA's broad applicability beyond classification (SSL/InfoNCE, NLP).
- Last but not least, we discussed with **Reviewer tLzt** regarding baselines (added curvature, MIR/HDR, etc.), the necessity of kurtosis, and statistical significance. Moreover, we conducted additional ablations clarifying the dynamics of sampling bias and train/validation splits, clarifying the relationship between GWA and validation sets under specific conditions.
- Unfortunately, **Reviewer RSWb** remained unresponsive throughout the discussion period, despite our rebuttal addressing their questions on applicability (SSL, NLP, optimizer independence) and despite reminders from both the authors and the AC. This prevented any dialogue to ensure their concerns were fully resolved.

We once again thank you all for your time and valuable feedback!

-- The Authors of Paper 4085

---

### Decision · Program_Chairs · 2025-09-17

**Decision:**

Accept (poster)

**Comment:**

This paper proposes an efficient per-sample estimator of gradient–parameter alignment, focusing on last-layer gradients computed in batches. The method provides a practical signal for early stopping and identifying influential training samples.

Reviewers raised concerns about broader evaluation across tasks, additional ablations on metric design, comparisons with related work, more experimental evidence, and new theoretical analysis. The authors actively engaged in the discussion and addressed most concerns, leading to one Accept, two borderline accepts, and two borderline rejects.

The AC carefully considered the two borderline rejects: Reviewer tLzt’s concerns were largely addressed, while Reviewer RSWb remained unresponsive during the discussion. The remaining weaknesses involve evaluation on additional tasks/settings and providing new theoretical justification, which are valuable but largely beyond the empirical scope of this work. Given that the proposed metric is simple, efficient, and effective, with potential impact on future research regarding generalization, the AC recommends acceptance of this paper and encourages the authors to include more evaluation on other tasks/settings in the final version.